# Lower-stratospheric aerosol measurements in eastward shedding vortices over Japan from the Asian summer monsoon anticyclone during the summer of 2018

Masatomo Fujiwara[1], Tetsu Sakai[2], Tomohiro Nagai[2], Koichi Shiraishi[3], Yoichi Inai[4,5], Sergey Khaykin[6], Haosen Xi[7], Takashi Shibata[8], Masato Shiotani[9], and Laura L. Pan[10]

[1] Faculty of Environmental Earth Science, Hokkaido University, Sapporo, 060-0810, Japan
[2] Meteorological Research Institute, Japan Meteorological Agency, Tsukuba, 305-0052, Japan
[3] Faculty of Science, Fukuoka University, Fukuoka, 814-0180, Japan
[4] Graduate School of Science, Tohoku University, Sendai, 980-8578, Japan
[5] Now at Japan Meteorological Agency, Sapporo, 060-0002, Japan
[6] LATMOS/IPSL, UVSQ, Sorbonne Université, CNRS, Guyancourt, 78280, France
[7] Graduate School of Environmental Science, Hokkaido University, Sapporo, 060-0810, Japan
[8] Graduate School of Environmental Studies, Nagoya University, Nagoya, 464-8601, Japan
[9] Research Institute for Sustainable Humanosphere, Kyoto University, Uji, 611-0011, Japan
[10] National Center for Atmospheric Research, Boulder, CO 80301, USA

*Correspondence to*: Masatomo Fujiwara (fuji@ees.hokudai.ac.jp)

## Abstract

Eastward airmass transport from the Asian summer monsoon (ASM) anticyclone in the upper troposphere and lower stratosphere (UTLS) often involves eastward shedding vortices, which can cover most of the Japanese archipelago. We investigated the aerosol characteristics of these vortices by analysing data from two lidar systems in Japan, at Tsukuba (36.1°N, 140.1°E) and Fukuoka (33.55°N, 130.36°E), during the summer of 2018. We observed several events with enhanced particle signals at Tsukuba at 15.5–18 km altitude (at or above the local tropopause) during August–September 2018, with a backscattering ratio of ~1.10 and particle depolarization of ~5% (i.e., not spherical, but more spherical than ice crystals). These particle characteristics may be consistent with those of solid aerosol particles, such as ammonium nitrate. Each event had a timescale of a few days. During the same study period, we also observed similar enhanced particle signals in the lower stratosphere at Fukuoka. The upper troposphere is often covered by cirrus clouds at both lidar sites. Backward trajectory calculations for these sites for days with enhanced particle signals in the lower stratosphere and days without indicate that the former airmasses originated within the ASM anticyclone, and the latter more from edge regions. Reanalysis carbon-monoxide and satellite water-vapour data indicate that eastward shedding vortices were involved in the observed aerosol enhancements. Satellite aerosol data confirm that the period and latitudinal region were free from the direct influence of documented volcanic eruptions and high latitude forest fires. Our results indicate that the Asian Tropopause Aerosol Layer (ATAL) over the ASM

region extends east towards Japan in association with the eastward shedding vortices, and that lidar systems in Japan can detect at least the lower stratospheric portion of the ATAL during periods when the lower stratosphere is undisturbed by volcanic eruptions and forest fires. The upper tropospheric portion of the ATAL is either depleted by tropospheric processes (convection and wet scavenging) during eastward transport or is obscured by much stronger cirrus cloud signals.

# 1 Introduction

The Asian Summer Monsoon (ASM) circulation includes a continental-scale anticyclone centred over the Tibetan Plateau, spanning from the Middle East to East Asia in the upper troposphere and lower stratosphere (UTLS). Satellite observations show elevated levels of trace gases of surface origin (e.g., Randel et al., 2010; Santee et al., 2017), aerosol particles (e.g., Vernier et al., 2015, 2018), and water vapour (e.g., Randel et al., 2015; Santee et al., 2017) within the ASM anticyclone due to active convection in this region and season. The ASM anticyclone exhibits distinct sub-seasonal variability due to westward and eastward shedding vortices (e.g., Popovic and Plumb, 2001; Amemiya and Sato, 2018), with the latter possibly being dynamically linked to the Bonin (or Ogasawara) High in the western Pacific (Enomoto et al., 2003) and constituting a major transport pathway of ASM airmasses to the whole Northern Hemisphere (NH) midlatitude UTLS through the westerly jet stream (e.g., Garny and Randel, 2013; Vogel et al., 2014, 2016; Ungermann et al., 2016; Pan et al., 2016; Fadnavis et al., 2018; Luo et al., 2018; Honomichl and Pan, 2020). Eastward shedding vortex events occur once in every 10–20 days during the NH summer, with a horizontal scale of 20°–30° longitude (2000–3000 km), and with a few days to one week of influence over the Japanese archipelago (e.g., Honomichl and Pan, 2020).

The enhanced aerosol particle signature in the ASM anticyclone at 14–18 km altitude was first discovered from satellite observations (Vernier et al., 2011) and thereafter referred to as the Asian Tropopause Aerosol Layer (ATAL). It was later verified from in situ balloon-borne measurements (Vernier et al., 2015, 2018; Yu et al., 2017; Brunamonti et al., 2018; Hanumanthu et al., 2020). Information on the chemical composition of the ATAL particles is limited (e.g. Martinsson et al., 2014; Vernier et al., 2018; Höpfner et al., 2019). Based on model simulations, the ATAL is expected to consist of carbonaceous and sulphate materials, mineral dust, and nitrate particles (e.g., Fadnavis et al., 2013; Gu et al., 2016; Lau et al., 2018; Fairlie et al., 2020; Bossolasco et al., 2020). Through analysis of satellite and high-altitude aircraft observations and laboratory experiments, Höpfner et al. (2019) provided evidence that a considerable part of the ATAL may contain solid ammonium nitrate ($NH_4NO_3$) particles. Their satellite data analysis using Cryogenic Infrared Spectrometers and Telescopes for the Atmosphere (CRISTA) data indicates enhanced $NH_4NO_3$ signals around the tropopause, both in the ASM region and the western Pacific (including Japan) during 8–16 August 1997 (with the western Pacific signals suggestive of shedding vortices); also, their analysis of satellite Michelson Interferometer for Passive Atmospheric Sounding (MIPAS) data together with CRISTA data show that the mass of $NH_4NO_3$ in the ASM region at 13–17 km peaks around August. It is also noted that Vernier et al. (2015, in their Figure 2b) showed mean eastward extension of the ATAL to the Japanese archipelago by averaging Cloud–

Aerosol Lidar with Orthogonal Polarization (CALIOP) data for July–August 2006–2013, although the role of synoptic disturbances, such as eastward shedding vortices, in the ATAL eastward extension does not appear to have been investigated using CALIOP data.

The "westward" extension of the ATAL to northern midlatitudes was reported by Khaykin et al. (2017), based on ground-based lidar at the Observatoire de Haute-Provence (OHP) in southern France (43.9°N, 5.7°E), with a layer of enhanced aerosol in the lower stratosphere with an average backscattering ratio (BSR; related to particle size and density) value of 1.05 being a systematic feature during August–October. This aerosol layer was shown to correlate with the seasonal water-vapour maximum, suggesting the influx of convectively moistened air from the ASM anticyclone, whose influence on the extratropical lower
stratosphere in late summer to early winter is well known (e.g., Vogel et al., 2014; Müller et al., 2016; Rolf et al., 2018).

Some lidar systems currently in operation in Japan are capable of measuring UTLS aerosol characteristics, including those at the Meteorological Research Institute (MRI), Tsukuba (36.1°N, 140.1°E; Sakai et al., 2016) and Fukuoka University, Fukuoka (33.55°N, 130.36°E; Yasui et al., 1995). Both systems measure the BSR and particle depolarization ratio (PDR; related to the
degree of particle non-sphericity). Previous studies using data from these systems investigated the impacts of the large-scale tropical volcanic eruptions and other recent eruptions (Uchino et al. 1993; Sakai et al. 2016), and spring-time transport of dust particles from the Asian continent, "Kosa" events (yellow sand/dust events) (Sakai et al. 2003), amongst others; however, the data have not been investigated extensively for the possible detection of the ATAL from ASM circulation, i.e., its "eastward" extension, partly because extensive summer-time cloud cover often prevents lidar sensing of the UTLS region, and partly
because ATAL signals are much weaker than volcanic signals. In this paper, focusing on the July–September 2018 period, we investigate whether these lidars are capable of measuring ATAL signals associated with eastward shedding vortices from the ASM anticyclone, with combined analyses of backward trajectories, chemical reanalysis data, and satellite data for full understanding of the lidar observations. The remainder of this paper is organised as follows. Section 2 describes the lidar and other data analysed in this paper. Section 3 presents the results and discussion, and Section 4 concludes the findings.

## 2 Data description

### 2.1 Lidar data

The lidar system at the MRI, Tsukuba (36.1°N, 140.1°E) used in this study is an Nd:YAG system operated at a wavelength of 532 nm, with a capability of both BSR and PDR measurements (Sakai et al., 2016), and which has been operating continuously
since 2002. We define PDR as $S/P$, where $S$ and $P$ are the background-subtracted lidar photon counts of the perpendicular ("senkrecht" in German) and parallel components, respectively, with respect to the polarization plane of the emitted laser light. The temporal and height resolutions of the original processed data are 5 min and 7.5 m, respectively. Quality control has been

done primarily to flag data points influenced by thick cloud layers. To obtain vertical profiles of BSR and PDR with high signal-to-noise ratios, data were averaged over 150 m and 3 h, with time intervals of 18–21, 21–00, 00–03, and 03–06 local time (LT) for the use in this paper. BSR data were normalised to unity at the 30–33 km altitude where aerosol backscattering is assumed to be negligible, and PDR values were obtained using the method of Adachi et al. (2001).

The lidar system at Fukuoka (33.55°N, 130.36°E) used in this study is also an Nd:YAG system operated at a wavelength of 532 nm, with PDR measurement capability. This system has been operated manually only during nights under clear-sky/non-rainy conditions; during July–September 2018, the system was operated on 11 nights. Vertical profiles were averaged over 900 m and 4 h for each night for the use in this paper. The PDR for Fukuoka is originally defined as $S/(P+S)$, which has been converted to $S/P$ for this paper.

The uncertainties of lidar data discussed here are applicable to both systems. The BSR uncertainties were estimated as follows. The random component was estimated from the photon counts of the backscatter signals at 532 nm after temporal and vertical averaging by assuming Poisson statistics. Other sources of BSR uncertainties (biases) were estimated by assuming the uncertainty of the normalization value of BSR to be $8.5 \times 10^{-3}$ (Russell et al., 1979, 1982) and that of the extinction-to-backscatter ratio to be 30 sr (Jäger and Hofmann, 1991; Jäger et al., 1995). The total uncertainty of BSR was then estimated to be 2–3 % typically around the tropopause. The PDR uncertainties were estimated from the parallel and perpendicular components of backscatter signals at 532 nm. Other sources of PDR uncertainties (biases) include (1) the uncertainty in calibration of the total depolarization ratio (TDR), due to both particles and air molecules, and (2) the BSR uncertainty. Uncertainty (1) was estimated as follows. In the TDR calibration (Adachi et al., 2001), we subtracted depolarization caused by the lidar system (DEPsys) estimated from the observed TDR and BSR obtained in the altitude region where aerosol backscattering is negligible (i.e., BSR equals unity, and TDR equals the molecular depolarization ratio), or where spherical particles predominate (i.e., in lower tropospheric water clouds). DEPsys errors result in PDR bias. For example, a DEPsys error of ±0.2% results in a ±2% bias in PDR where BSR = 1.1. Uncertainty (2) arises mainly from our assumption that aerosol backscattering is negligible at 30–33 km altitude. We also assumed an aerosol extinction-to-backscatter ratio of 50 sr over the whole measurement height range. These assumptions result in errors in BSR and thus PDR. For example, BSR errors of +0.05 and –0.05 result in a bias of –1% and +3% in PDR, respectively, where BSR = 1.1 and TDR = 0.7%. Based on these considerations, we estimate that the total PDR uncertainty (random plus bias errors) is ≤±5% PDR.

## 2.2 Other data

Backward trajectories are calculated using the trajectory model used by Inai (2018) and Inai et al. (2018) and the most recent global atmospheric reanalysis dataset by the European Centre for Medium-Range Weather Forecasts (ECMWF), ERA5 (Hersbach et al., 2020), with 37 pressure levels up to 1 hPa and horizontal and temporal resolutions of 0.25° × 0.25° and 1 h,

respectively. ERA5 temperature data in the tropical tropopause layer have been evaluated by Tegtmeier et al. (2020). Lagrangian transport calculations using ERA5 and its predecessor ERA-Interim have been compared by Hoffmann et al. (2019) and Li et al. (2020).

The Copernicus Atmosphere Monitoring Service (CAMS) atmospheric-composition reanalysis dataset produced by the ECMWF (Inness et al., 2019) is used to analyse signatures of the ASM anticyclone and its eastward shedding vortices, with 25 pressure levels up to 1 hPa and horizontal and temporal resolutions of $0.75° \times 0.75°$ and 3 h, respectively. Carbon monoxide (CO), temperature ($T$), and geopotential ($\Phi$) data are primarily analysed in this paper. CO is chosen because it is a good tracer for polluted air of surface origin (e.g., Luo et al., 2018). Although CO and ATAL aerosol particles do not necessarily have the

same emission sources, CO is a good chemical tracer to indicate the location of the ASM anticyclone. CO data on pressure levels are projected onto isentropic surfaces using temperature data. In the CAMS, the Measurement of Pollution in the Troposphere (MOPITT) thermal infrared (TIR) satellite total-column CO data are assimilated, but Microwave Limb Sounder (MLS) and Infrared Atmospheric Sounding Interferometer (IASI) CO data are not. CAMS CO data are originally provided in mass mixing ratio, kg kg$^{-1}$, which are converted to volume mixing ratio, ppbv, for this study. It is noted that a quick comparison

(not shown) with MLS Version 4.2 Level 2 CO data (Santee et al., 2017; Livesey et al., 2020) at 400 K isentropic surface (in the form of longitude-time diagram like the one in Section 3.2) shows that CAMS CO data are roughly ~10 ppbv greater than MLS CO over Japan during August–September 2018, but also shows that eastward extension signals coming over Japan agree fairly well qualitatively within the differences in spatio-temporal sampling of the two datasets. The CAMS dataset also includes different types of aerosol particles, but they are not included in this study because relevant variables such as aerosol BSR and

NH$_4$NO$_3$ concentration are not included. Montgomery streamfunction (MSF), defined as MSF = $c_p T + \Phi$, where $c_p$ is specific heat of dry air at constant pressure, in isentropic coordinates corresponds to geopotential (height) in isobaric coordinates (e.g., Popovic and Plumb, 2001; Santee et al., 2017; Amemiya and Sato, 2018; Salby, 1996), and thus is a good dynamical indicator of the ASM anticyclone. Potential vorticity (PV) on isentropic surfaces (e.g., at 360–380 K) is often used as a dynamical tracer in studies of the ASM anticyclone (e.g., Popovic and Plumb, 2001; Garny and Randel, 2013; Ploeger et al., 2015; Amemiya

and Sato, 2018); however, PV at and above 400 K (the isentropic surface we will focus in Section 3.2) is not very useful to analyse the ASM anticyclone boundary. Thus, we will analyse MSF at 400 K surface calculated from CAMS data.

MLS Version 4.2 Level 2 water-vapour data (Santee et al., 2017; Livesey et al., 2020) are analysed because water vapour is also a good tracer of the ASM anticyclone. We use MLS data rather than CAMS data for lower stratospheric water vapour

because MLS data have been well validated with e.g., balloon-borne frost-point hygrometers (e.g., Hurst et al., 2016; Fujiwara et al., 2010; Vömel et al., 2007), while reanalysis water vapor data are in general less reliable in the lower stratosphere (e.g., Davis et al., 2017). We found (not shown) that CAMS water vapour volume mixing ratio data (converted from the original specific humidity data) are greater than MLS data at 400 K isentropic surface over Japan during July–September 2018 (e.g., the differences are roughly ~2 ppmv for the wet signals around the longitudes of Japan in August 2018).

The possible influence of volcanic eruptions and wildfire events is investigated using two satellite aerosol-particle datasets, one providing vertical extinction profile data at 675 nm from the Ozone Mapping and Profiler Suite (OMPS) Limb Profiler (LP), Level 2 Version 1.5 (Chen et al., 2018), and the other attenuated scattering-ratio data from the CALIOP onboard the Cloud–Aerosol Lidar and Infrared Pathfinder Satellite Observation (CALIPSO) satellite (Thomason et al., 2007; Winker et al.,

2007, 2010). CALIOP Level 3 monthly-mean stratospheric aerosol data (CAL_LID_L3_Stratospheric_APro-Standard-V1-00) are used in this study; in this data product, clouds and Polar Stratospheric Clouds (PSCs) have been removed based on the information of particulate extinction-to-backscatter (lidar) ratio and the multiple-scattering factor profile (Young and Vaughan, 2009; Kim et al., 2018; https://www-calipso.larc.nasa.gov/resources/calipso_users_guide/data_summaries/l3/lid_l3_stratospheric_apro_v1-00_v01_desc.php).


## 3 Results and discussion

### 3.1 Lidar measurements

Time–height distributions of BSR and PDR observed in the UTLS at Tsukuba are shown in Figure 1, and the corresponding vertical profiles are shown in Figure 2. Because the PDR has more missing data points, the TDR time–height distribution is

also shown in Figure A1. Days with missing data (white regions; Fig. 1) are due to thick summer-time rain clouds in the lower-to-middle troposphere, which prevented the laser light reaching the middle stratosphere. However, some events with enhanced particle signals are evident just above the tropopause at 15.5–18 km, and last for a few days, mainly in August, but with some in September. In particular, the event peaking at around 21 August and spanning 18–26 August was the strongest one among those the lidar successfully measured during the three-month period. We also observe another strong event

around 9 August at 15–17 km, although missing observations before and after this date prevent characterisation of the temporal scale of the event; furthermore, the tropopause height was highly variable at the time and was located at 17 km on that date, situating the aerosol enhanced layer temporarily in the troposphere. Fig. 2 shows that enhanced particle signals at 15.5–18 km were often observed in August, sometimes in September, but not in July. Typical BSR and PDR values of enhanced signals are ~1.10 (1.07–1.18) and ~5% (3%–10%), respectively (Figs. 1 and 2). Below the tropopause, strong

signals were sometimes recorded with BSR values of >1.25 and with PDR values >>10%. In general, the PDR values are 0% for spherical particles (i.e., water clouds in the troposphere and liquid $H_2SO_4$ particles in the stratosphere) and are >25%–30% for ice cirrus particles (e.g., Sakai et al., 2003; Fujiwara et al., 2009). Strong signals in the upper troposphere are thus due to ice cirrus clouds. Enhanced signals in the lower stratosphere (15.5–18 km) may be due to solid particles, as indicated by the PDR values of ~5% (3%–10%). Taking PDR uncertainties (Sect. 2.1) into account, these values can be considered as

small, but non-zero, values. The PDR values of these signals, together with the region being above the local tropopause in most cases, strongly suggest that they are not ice cirrus particles. However, the possibility of a mixture of spherical $H_2SO_4$

particles (i.e., background stratospheric sulphate particles) and highly non-spherical particles, such as ice, volcanic ash (Prata et al., 2017), and wildfire smoke (Haarig et al., 2018), cannot be precluded only with our lidar data. We will come back to this issue in Sect. 3.3 after investigating several other data. Before looking at the Fukuoka results, it is noted that for Tsukuba we do not plot the data with "relative" uncertainty of PDR larger than 30%; this treatment resulted in removing data points with BSR values lower than ~1.05 where background spherical sulphate particles (with PDR values of <2%) were presumably predominant.

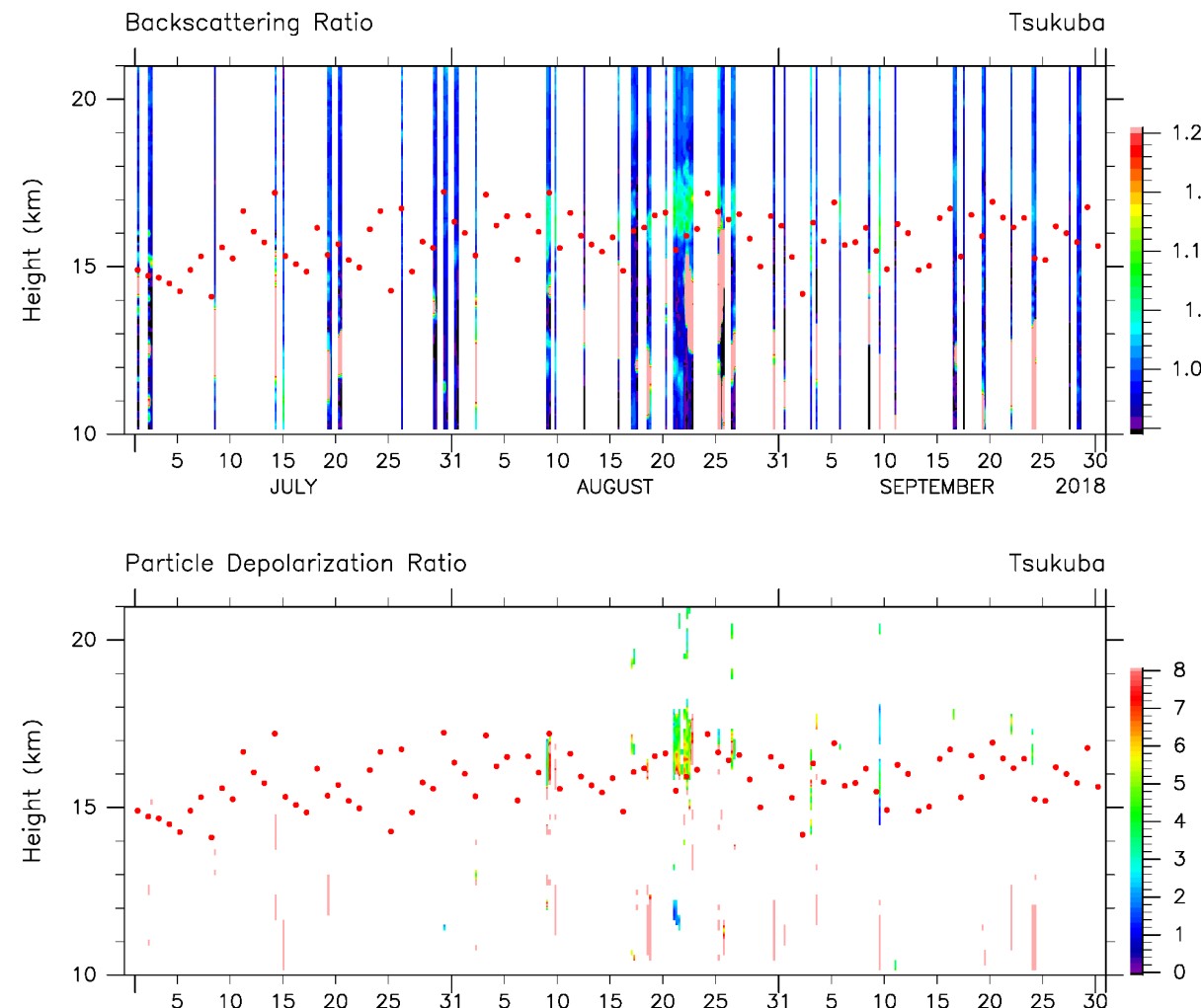


**Figure 1** Time–height distributions of (top) backscattering ratio and (bottom) particle depolarization ratio (%) during July–September 2018, as measured using the lidar system at MRI, Tsukuba. For each day, four time slots (i.e., 18–21, 21–00, 00–03, and 03–06 LT) are prepared, with 3-h averaged data filling the slots where thick lower-to-middle tropospheric clouds do not exist. Red dots indicate the daily
(first) lapse-rate tropopause locations determined by the Japan Meteorological Agency (JMA), based on 21 LT radiosonde data taken at the JMA "Tateno" site (which shares the same site as the MRI).

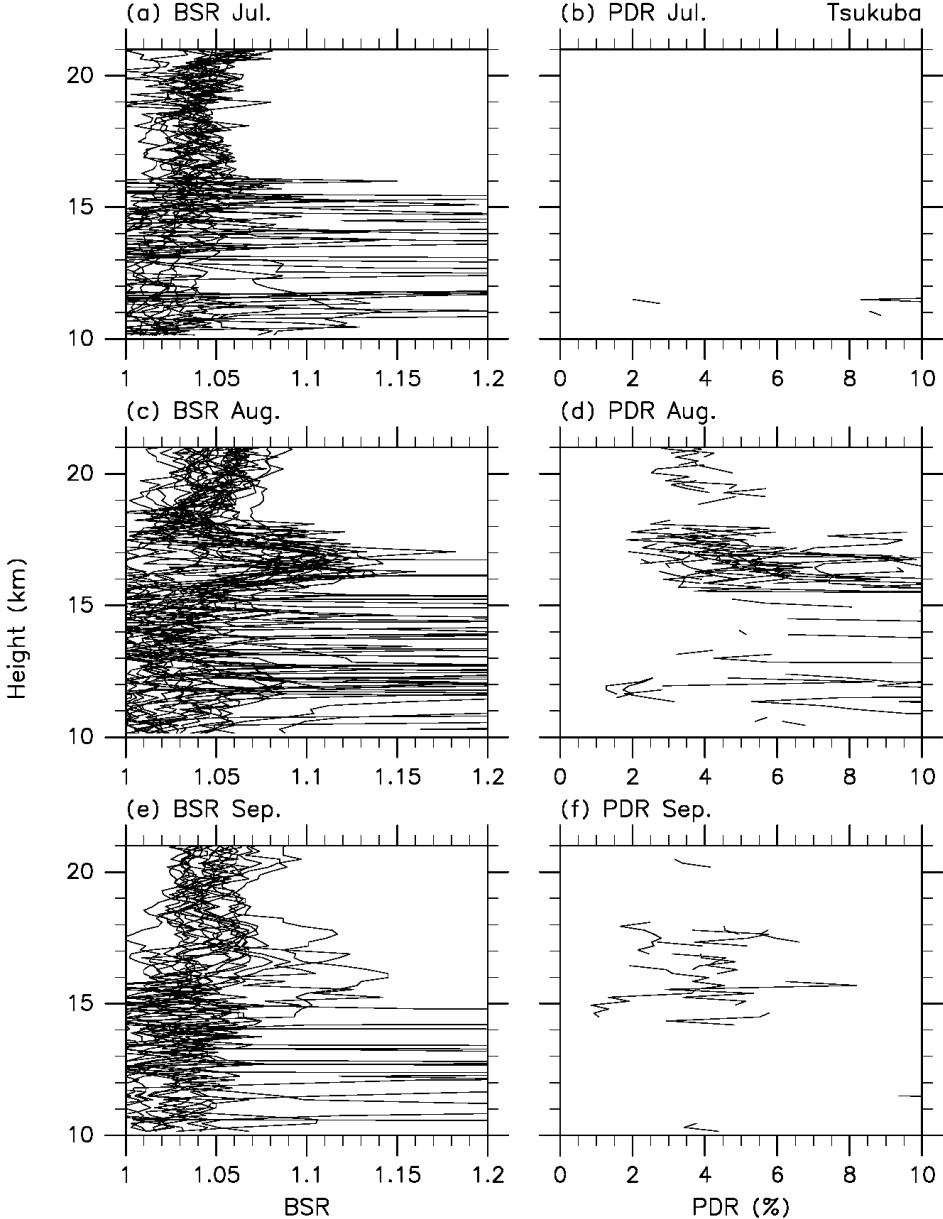


**Figure 2** Vertical profiles of (a, c, e) backscattering ratio (BSR) and (b, d, f) particle depolarization ratio (PDR, in %) in (a, b) July, (c, d) August, and (e, f) September 2018 obtained using the lidar system at MRI, Tsukuba. It is noted that strong and noisy signals in BSR below ~15.5 km are due to cirrus clouds.

Vertical profiles of BSR and PDR observed at Fukuoka for 11 clear-sky/non-rainy nights during July–September 2018 are shown in Figure 3. Again, enhanced particle signals were observed mainly in August above the tropopause at 15.5–18 km. The BSR values were in the range 1.09–1.14, with PDR values of 1%–3% which are smaller than those observed at Tsukuba. It should be noted that the dates of lidar operation at Fukuoka did not overlap those at Tsukuba when strong enhancement was observed above the tropopause (e.g., 9 August, 18–26 August, and 9 September), perhaps partly explaining the differences between Figs. 2 and 3.

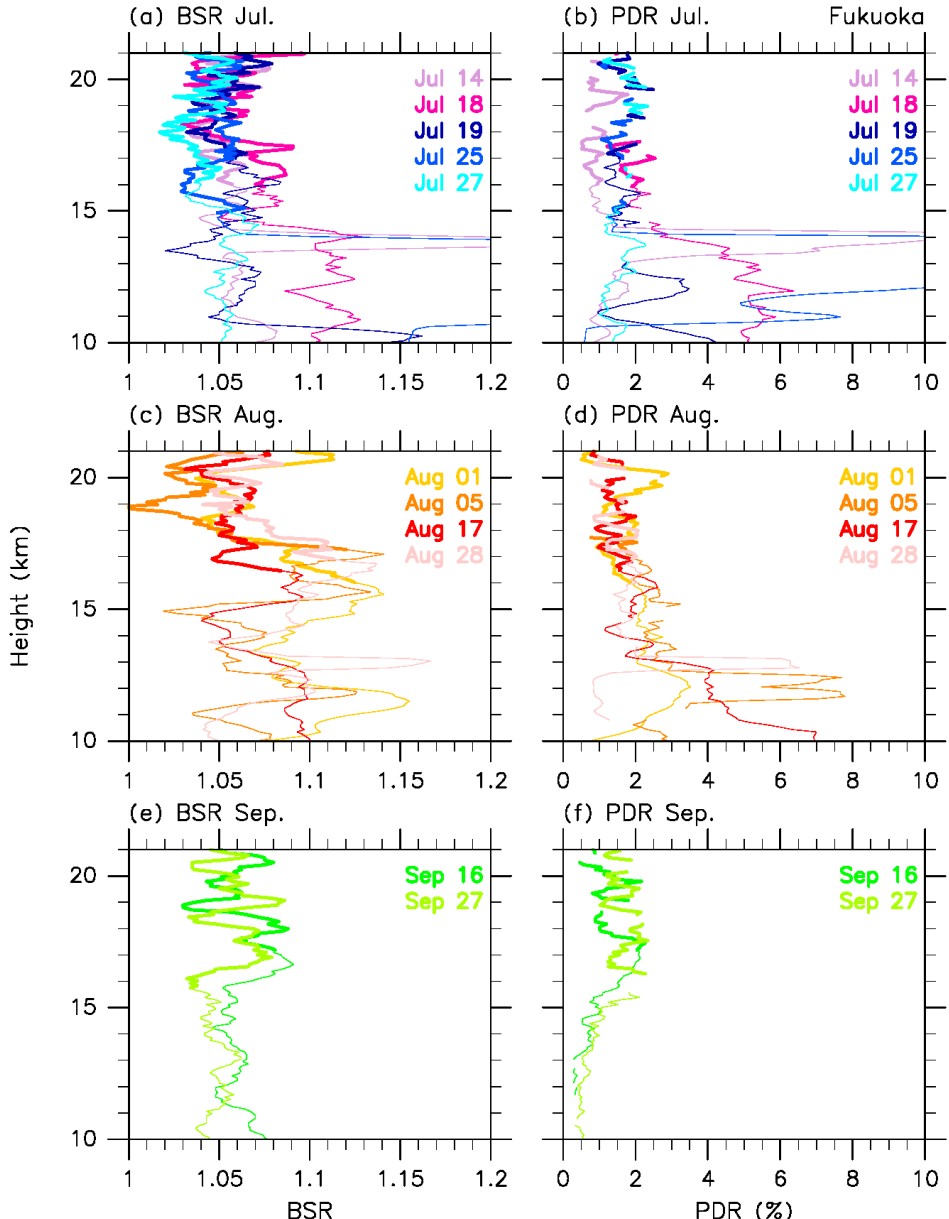


**Figure 3** Eleven vertical profiles of (a, c, e) backscattering ratio (BSR) and (b, d, f) particle depolarization ratio (PDR, in %) in (a, b) July, (c, d) August, and (e, f) September 2018 obtained using the lidar system at Fukuoka. Dates and colours are assigned in the legend where, for example, "Jul 14" refers to the night of July 14–15. Stratospheric portion of the profiles has been thickened using the daily (first) lapse-rate tropopause location information provided by the JMA, based on operational 21 LT radiosonde data taken at the JMA Fukuoka site (at

a ~4 km distance from the lidar site).

## 3.2 Trajectories and airmasses

Ten-day kinematic backward trajectories (using vertical wind) from Tsukuba and Fukuoka are shown in Figures 4 and 5,
respectively, with contrasting cases with or without aerosol particle enhancement in the 390–410 K potential-temperature
range (around 16.5–17.5 km at these stations). A potential temperature of 400 K corresponds to altitudes of ~17.1 km at
Tsukuba and ~17.3 km at Fukuoka, on average, during July–September 2018 (based on twice-daily radiosonde data at each
site, taken from http://weather.uwyo.edu/upperair/sounding.html), i.e., near the centre of the lower stratospheric BSR
enhancement. By comparing the results from Santee et al. (2017) with our own analysis, the 65 ppbv contours of monthly
mean CAMS CO data at 400 K potential temperature are chosen as an index of the boundaries of the ASM anticyclone (i.e.,
within the anticyclone, CO concentration is >65 ppbv). These trajectories indicate that airmasses over both stations come
mainly from the west, and sometimes via the north of Japan (indicative of the existence of vortices), and originated from the
ASM anticyclone well within 10 days. They also indicate that airmasses with enhanced aerosol particles at this height tend to
originate in regions within the ASM anticyclone at the altitudes 16.5–18 km, i.e., around or just below the tropopause, whereas
those without enhanced aerosol particles tend to originate from edge regions surrounding the anticyclone. Note that there is a
trajectory that originates in the Pacific south of Japan as low as 4 km (Figure 4, bottom, a small-scale spiral in purple); this is
associated with upward transport in the typhoon Soulik.

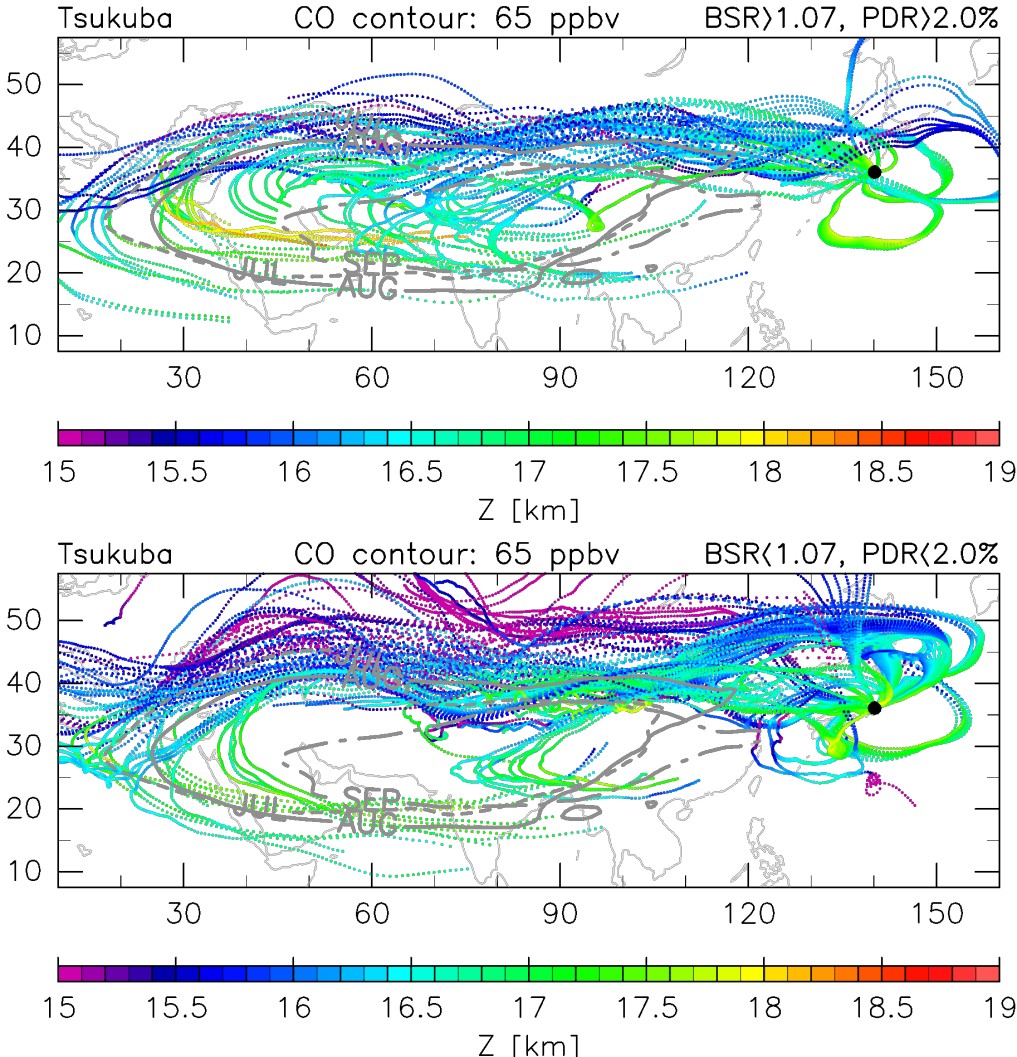

**Figure 4** Kinematic backward trajectories for 10 days starting from Tsukuba in the 390–410 K potential temperature range at 100 m geopotential height intervals on all days during July–September 2018 on which measurements were made, using ERA5 reanalysis data. Cases are sorted into two categories: (top) with and (bottom) without enhanced aerosol signals observed by lidar at the trajectory starting points. The conditions and number of trajectories for the former and the latter cases are, respectively, BSR > 1.07; PDR > 2.0%; 78 trajectories and BSR < 1.07; PDR <2.0%; 136 trajectories. Colours indicate geopotential height (Z) values of the trajectories. Grey contours indicate 65 ppbv monthly mean CAMS CO levels at 400 K potential temperature, roughly indicating monthly mean boundaries of the ASM anticyclone (dotted for July, solid for August, and dash-dotted for September).




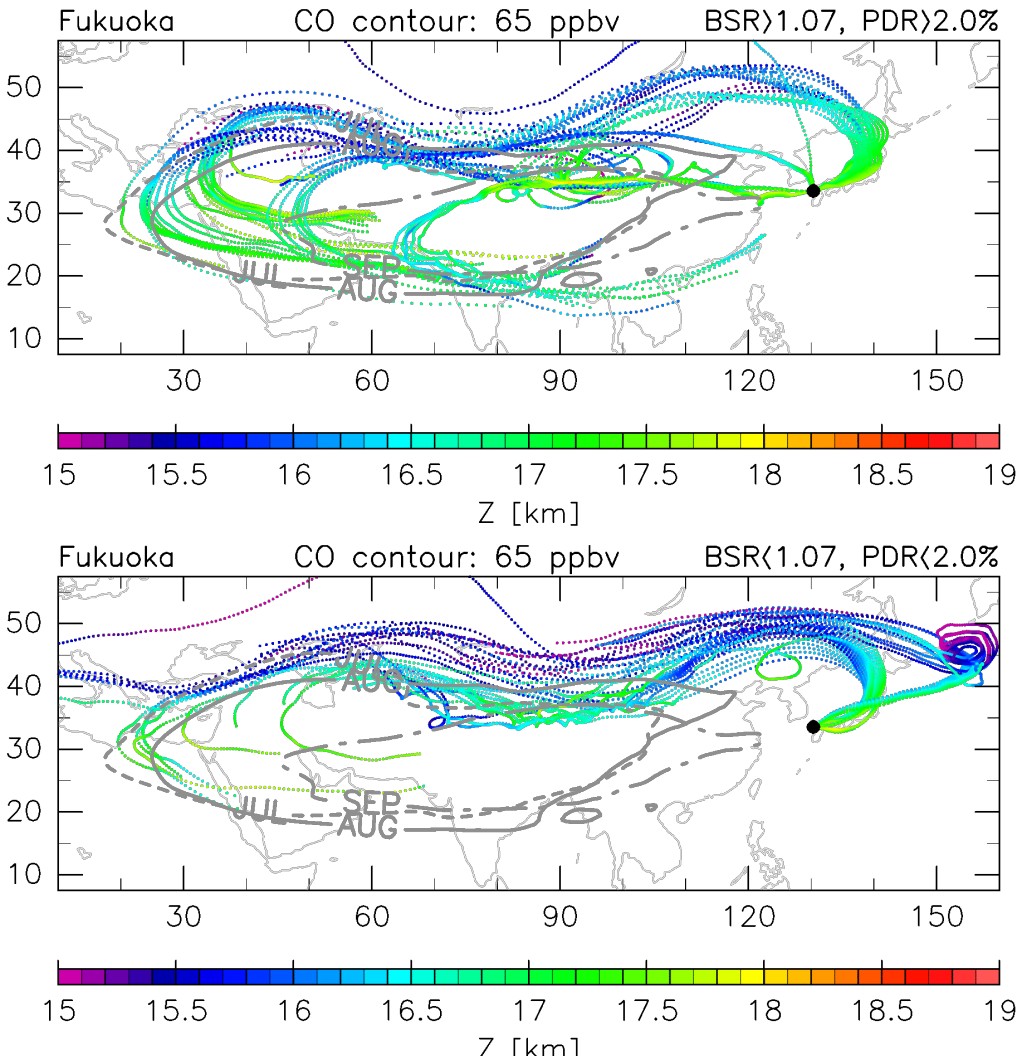

**Figure 5** As for Fig. 4, but for trajectories from Fukuoka, with (top) 44 and (bottom) 37 trajectories.


Horizontal distributions of CO and MSF at the 400 K potential-temperature surface during 18–23 August 2018 from the CAMS reanalysis data are shown in Figure 6. Again, note that a potential temperature of 400 K corresponds to ~17.1 km at Tsukuba and ~17.3 km at Fukuoka during July–September 2018. The distribution of MSF indicates the location of the ASM anticyclone from the dynamical viewpoint. The strongest particle signals during the three months were observed on 21 August in the lower stratosphere over Tsukuba. The airmass with high CO concentrations was transported eastward from the ASM anticyclone centred over the Tibetan Plateau (Fig. 6), with an anticyclonic vortex of ~20° longitude scale reaching the Japanese archipelago on 21 August, providing a clear signature of eastward shedding vortices from the ASM anticyclone (e.g., Luo et al., 2018). Daily averaged longitude–time CO distributions over 30°N–40°N are shown in Figure 7, with that latitude band chosen here because it includes the two lidar sites. The ASM anticyclone spans roughly 15°N–40°N, whereas the eastward shedding vortices are often located slightly to the north, at around 25°N–45°N, as indicated in Figs. 4–6; the latitude band must therefore be chosen carefully, depending on the research focus. In Fig.7, the 60-ppbv CO contour may be a good indicator of eastward shedding vortices. In July 2018, the eastward extension was weak, but in August there were three events that directly affected the two lidar sites, at 3–15, 20–24 (Fig. 6), and 28–31 August. In September, there were three events, at 3–8, 14–17, and 28–29 September. A comparison with Fig. 1 indicates that aerosol-particle enhanced events correspond relatively well to CO-enhanced events, although missing lidar data points (due to low-level clouds) result in the fact that only the 20–24 August event was relatively well observed, with the 3–15 August event being captured only on 9 August. The ASM anticyclone is also characterised as an airmass hydrated by active convection from below (e.g., Santee et al., 2017). The longitude–time distribution of MLS water vapour at 400 K, averaged over 30°N–40°N with 8°-longitude and 3-day bins, is shown in Figure 8. The water-vapour-enhanced events over Japan correspond well with the CO-enhanced events over the same region shown in Fig. 7.

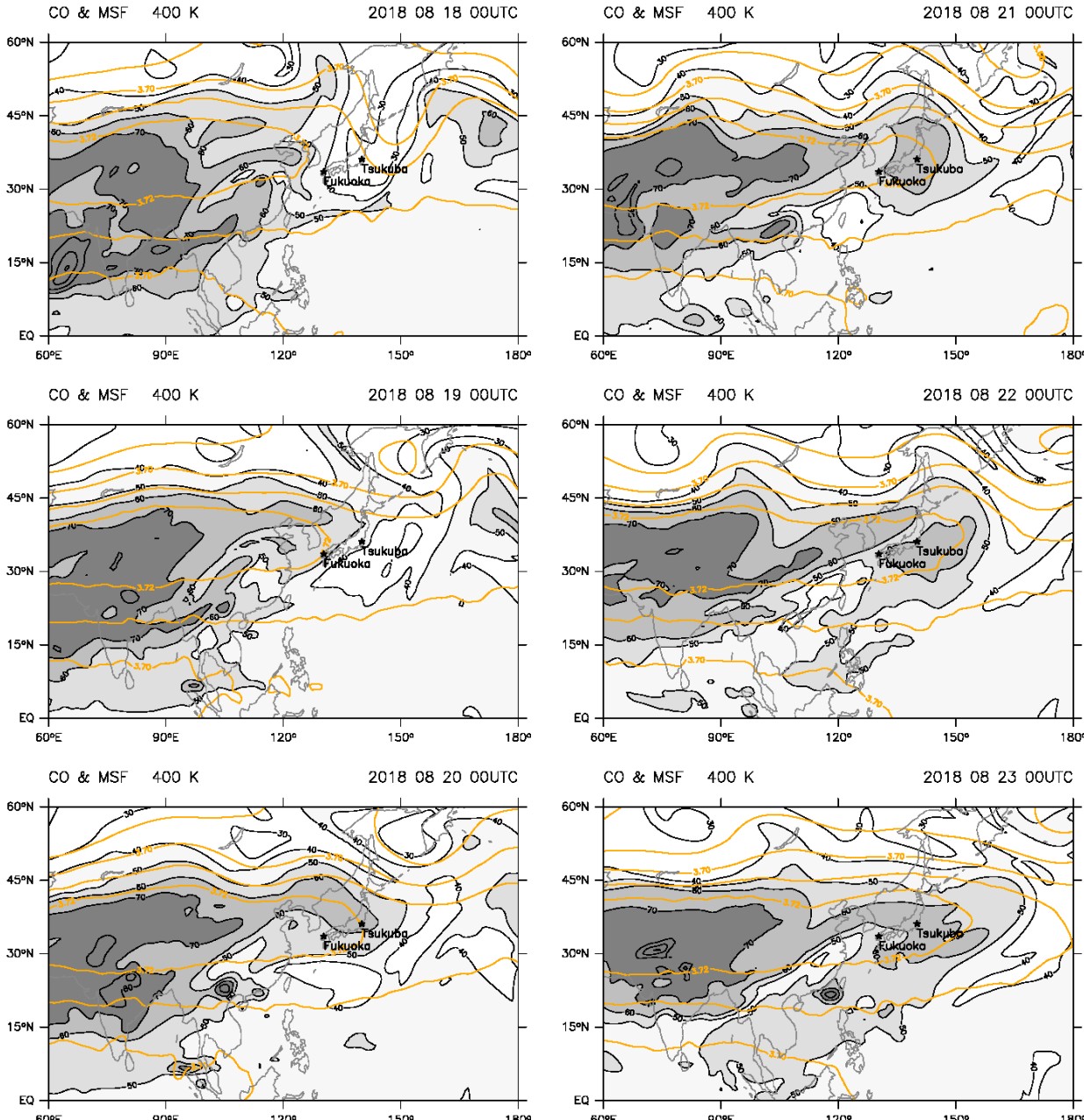

**Figure 6** Horizontal distribution of daily, instantaneous (00 UTC) CO (black contours with grey tone, with intervals of 10 ppbv) and Montgomery streamfunction (MSF; coloured contours at intervals of $0.01 \times 10^5$ m$^2$ s$^{-2}$) at the 400 K potential-temperature level during 18–23 August 2018 (dates indicated at top right of each plot), using CAMS reanalysis data.

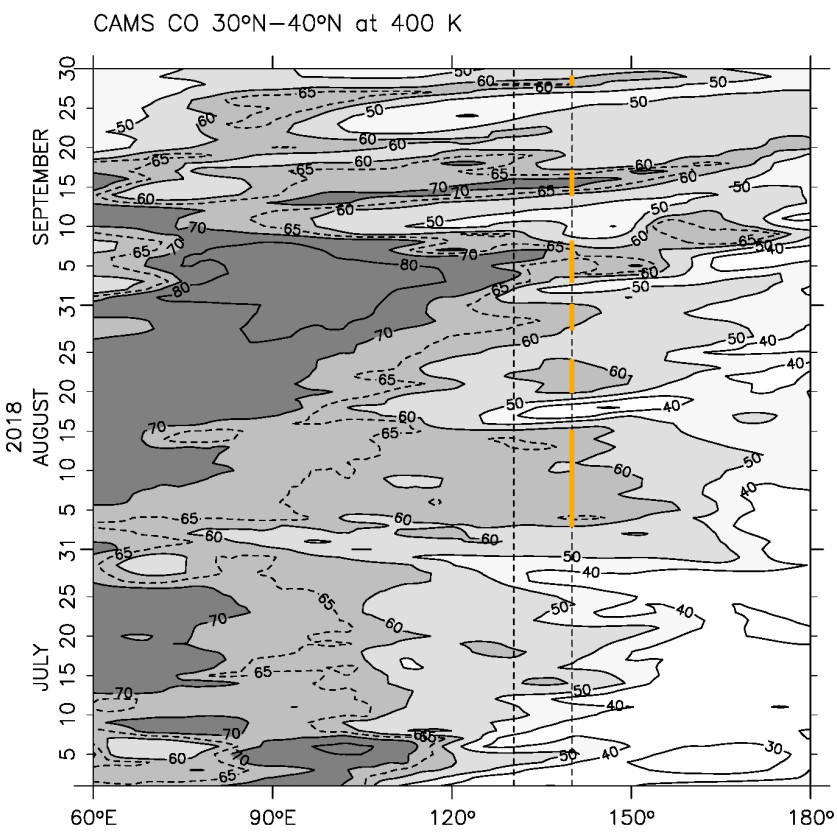

**Figure 7** Longitude–time distribution of daily averaged CO concentration at 400 K potential temperature averaged over 30°N–40°N, using CAMS reanalysis data. The contour interval is 10 ppbv, with 65 ppbv contours added (dotted). Vertical dotted lines indicate the locations of the two lidar sites, Fukuoka (130.36°E) and Tsukuba (140.1°E); furthermore, the periods along the longitude of Tsukuba when CO concentration was ≥~60 ppbv (i.e., 3–15, 20–24, and 28–31 August, and 3–8, 14–17, and 28–29 September) are shown as orange line segments.

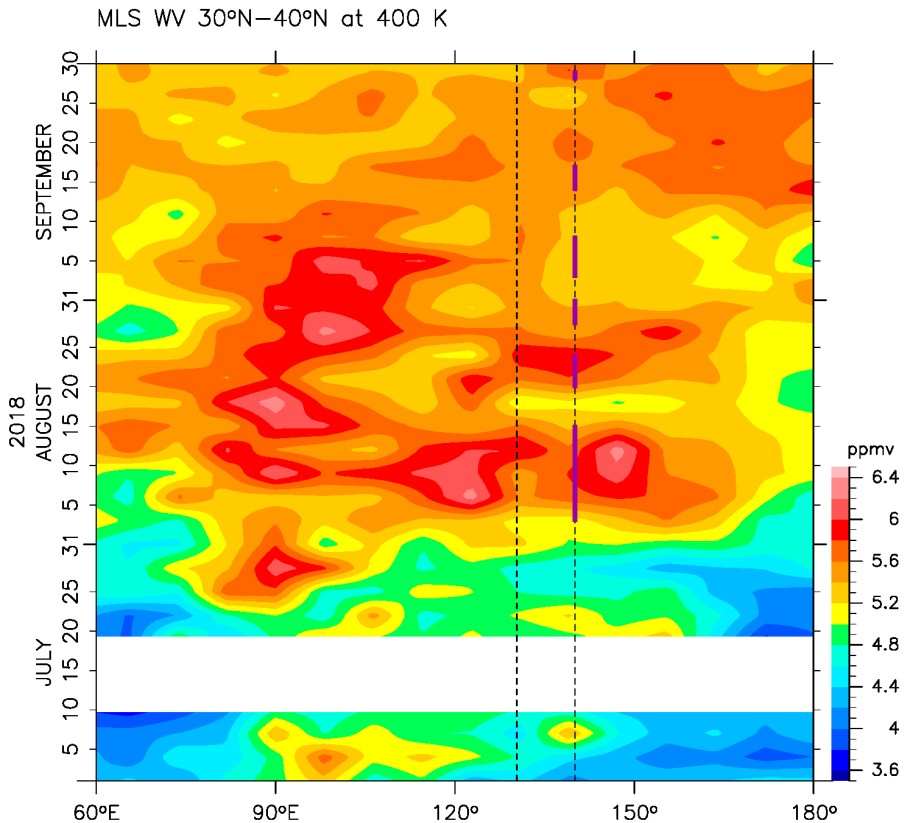

**Figure 8** As for Fig. 7, but for water vapour (in ppmv) at 400 K measured with the satellite MLS instrument. Data for the 30°N–40°N region have been aggregated into 3-day and 8°-longitude bins, each constituting about 10 individual data points. The white region indicates missing measurements. The purple line segments are the same ones but in orange in Fig. 7.

### 3.3 Investigation of other potential causes

Lidar is sensitive to various types of volcanic aerosol (e.g., Yasui et al., 1996; Sakai et al., 2016; Khaykin et al., 2017). The lower stratosphere is continuously influenced by volcanic eruptions (GVP, 2013), which inject various types of particles and gases into the atmosphere (e.g., Robock, 2000). Among them, solid ash particles may remain in the stratosphere for up to a few months, while liquid $H_2SO_4$ particles resulting from reaction of volcanic $SO_2$ and $H_2S$ gases with OH and $H_2O$ may remain for a year or more. Aerosol particles are also emitted from biomass burning and forest fires and, although these particles rarely

reach the stratosphere, extensive fire events can influence the stratospheric aerosol loading (e.g., Khaykin et al., 2018, 2020; Peterson et al., 2018; Kablick et al., 2020). In this section, the global lower-stratospheric aerosol loading during the summer of 2018 is investigated by the analysis of satellite aerosol data.

The time–latitude distribution of zonal-mean lower-stratospheric aerosol optical depth (AOD) at 675 nm from the OMPS LP

satellite instrument is shown in Figure 9. At high NH latitudes, the lower-stratospheric AOD increased in the summer of 2017 due to extensive wildfires in Canada (Khaykin et al., 2018; Peterson et al., 2018), but their influence became negligible by early 2018. In July 2018, the eruption of Ambae (or Aoba; 15.389°S, 167.835°E; GVP, 2019), Vanuatu, in the tropical western Pacific, caused increasing stratospheric AOD in the tropics. We also observed very weak signals around the same latitude from the beginning of April 2018, possibly due to the eruption of the same Ambae during March–April 2018 (GVP, 2018). However,

the lower stratospheric AOD at 25°N–40°N was relatively low during July–September 2018, at least on a zonal-mean scale. The monthly mean CALIOP attenuated scattering-ratio distribution due to aerosol particles at 17 km in July and August 2018 is shown in Figure 10 where the ATAL is evident, with enhanced aerosol signals over the ASM region. In August there was also a hint of eastward extension of the ATAL to Japan, with a slight increase in the scattering ratio. By August, effects of the Ambae eruption had extended to about half of the tropics, but had not reached Japan directly, at least not in a monthly mean

view (see also the 10-day backward trajectories; Figs. 4 and 5).

Finally, Chouza et al. (2020) showed that lidar measurements at Mauna Loa, Hawaii, indicated no signals from volcanic eruptions during the summer of 2018. Also, at the OHP lidar site in France (43.9°N, 5.7°E), no enhancement in the lower stratospheric aerosol abundance was observed during the summer of 2018. In summary, enhanced aerosol particle signals

observed at Tsukuba (36.1°N) and Fukuoka (33.55°N) were thus unlikely to be due to volcanic eruptions or northern wildfires.

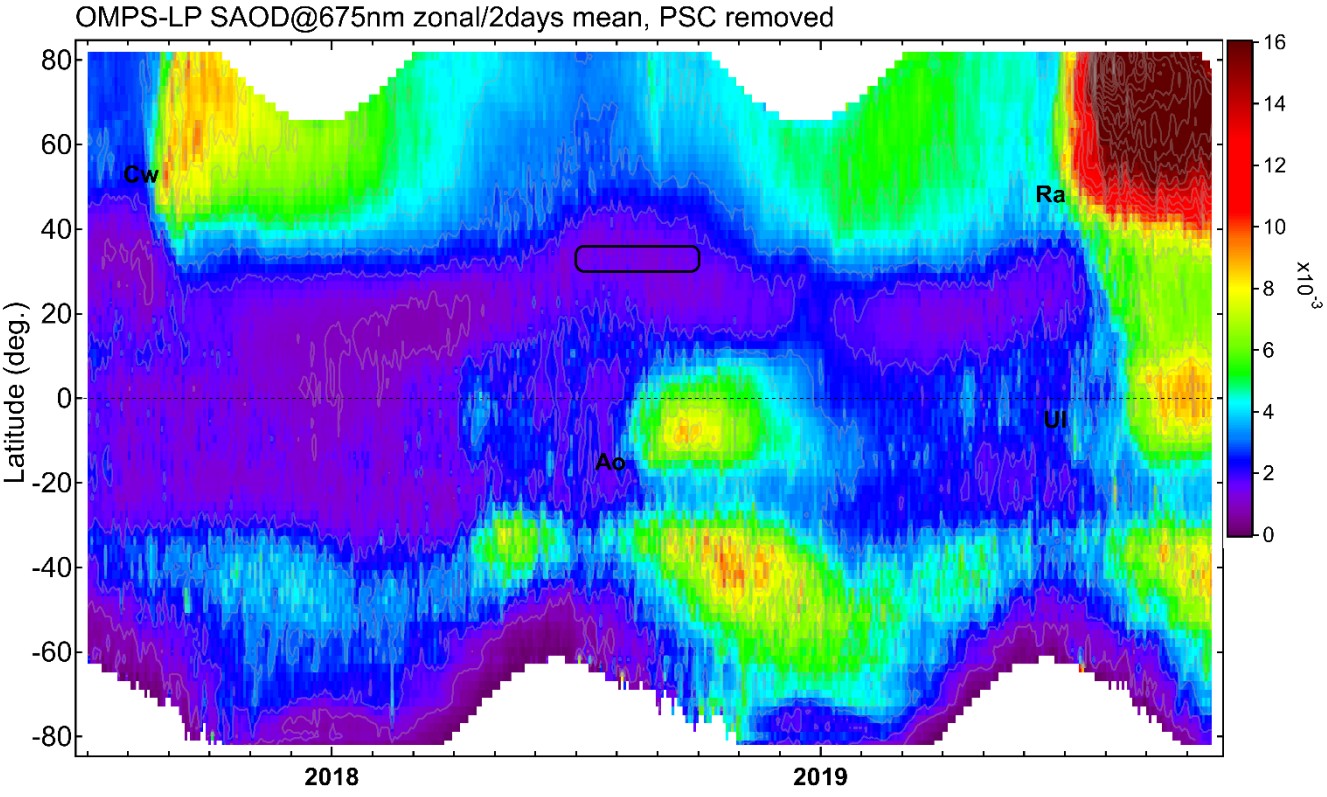


**Figure 9** Time–latitude distribution of zonal- and 2-day-mean lower-stratospheric aerosol optical depth at 675 nm between the tropopause and 21 km altitude, from July 2017 to October 2019, as calculated from OMPS LP satellite data. The tropopause altitude for each OMPS LP profile was provided within the OMPS LP dataset. Signals due to Polar Stratospheric Clouds (PSCs) have been removed. Major events that significantly enhanced NH stratospheric aerosol loading are labelled: Cw, Canadian wildfires in the summer of 2017; Ao, Ambae (Aoba) eruption, Vanuatu (July 2018); Ra, Raikoke eruption, Kuril Islands, Russia (June 2019); Ul, Ulawun eruption, Papua New Guinea (July, August, and October 2019) (GVP, 2013). The rectangular box indicates the period and location of the lidar measurements; white regions indicate missing measurements.



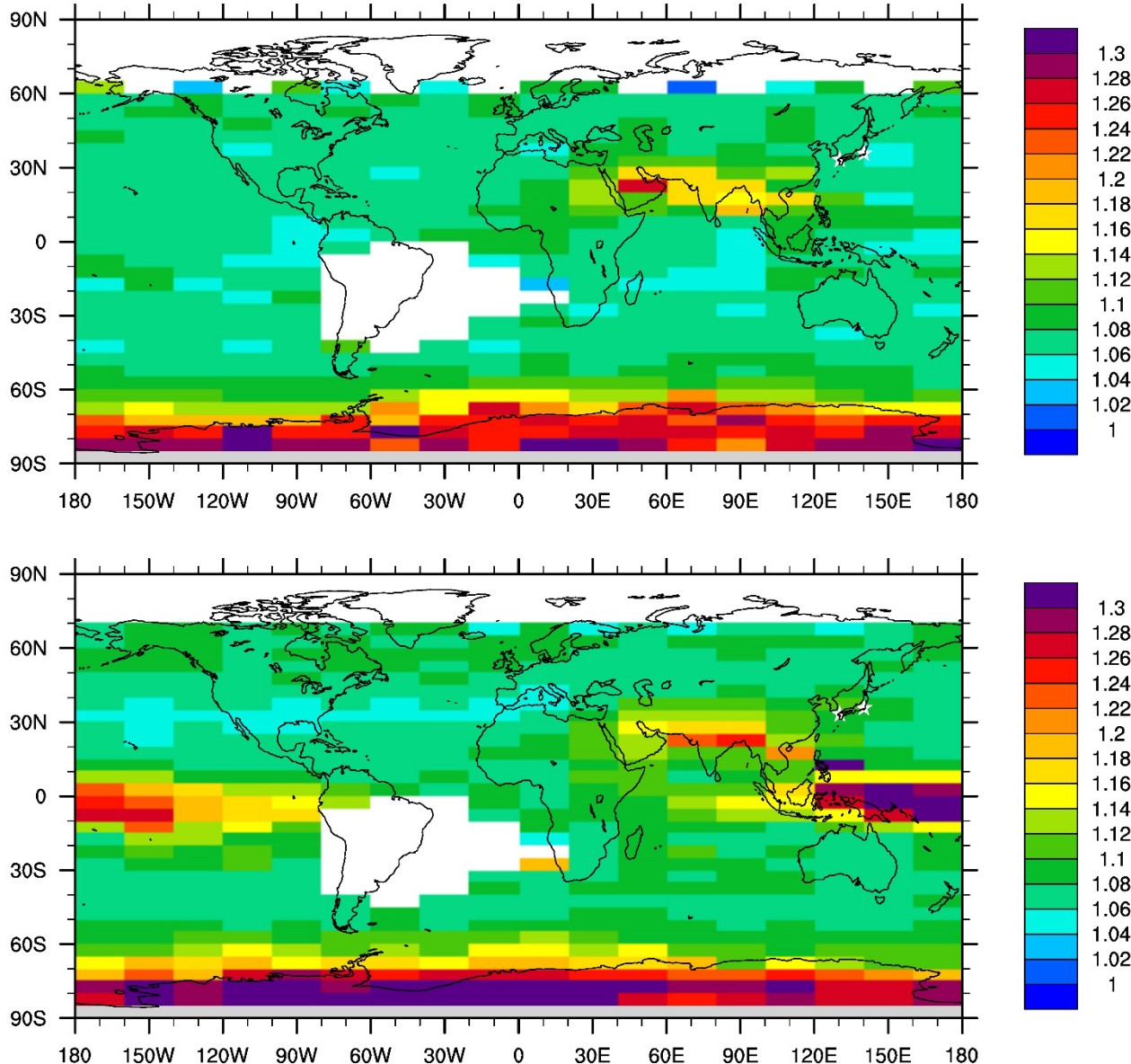

**Figure 10** Monthly mean horizontal distribution of attenuated scattering ratio at 17.042 km observed with the CALIOP satellite instrument in (top) July 2018 and (bottom) August 2018. Spatial bins are 5° in latitude, 20° in longitude, and 900 m in altitude. Clouds and PSCs have been removed (Sect. 2.2). It is noted that the CALIOP attenuated scattering ratio is defined as the ratio of the measured attenuated backscatter coefficients and the attenuated backscatter coefficients calculated from the molecular model, and its valid range is from 0.60 and 25.00. The two lidar station locations are marked with white stars. White regions indicate missing measurements. (See Noel et al. (2014) for the data missing over the South Atlantic region).


## 4 Summary and conclusions

Lidar aerosol-particle measurements made at Tsukuba and Fukuoka, Japan, during the summer of 2018 were investigated to determine whether these lidars are capable of detecting the eastward extension of the ATAL from the ASM anticyclone in the UTLS. Both lidars observed enhanced aerosol-particle signals between the local tropopause and up to a few km above it, with BSR values of ~1.10 (1.07–1.18) and PDR values of ~5% (3%–10%) at Tsukuba and with similar BSR but lower PDR values at Fukuoka. The PDR difference between the two sites may be due to the Fukuoka lidar operating on only 11 nights during the three-month period and due to the fact that the dates of lidar operation at Fukuoka did not overlap those at Tsukuba when strong enhancement was observed. The lidars often detected strong signals (BSR values of >1.25 and with PDR values >>10%) due to ice cirrus clouds below the tropopause. The Tsukuba measurements indicated that timescales of lower-stratospheric enhancements are a few days. Backward trajectory calculations and reanalysis CO data support the hypothesis that airmasses with enhanced aerosol signals originate in the ASM anticyclone and are transported over these sites in association with eastward shedding vortices. OMPS LP and CALIOP satellite data indicated that the 25°N–40°N region was not influenced by volcanic eruptions or extensive biomass burning events during July–September 2018. Our results indicate that the enhanced aerosol particle levels measured at Tsukuba and Fukuoka are due to eastward shedding vortices of the ATAL from the ASM anticyclone; i.e., they originated from pollutants emitted from Asian countries and transported vertically by convection in the ASM region.

The PDR values obtained at Tsukuba, i.e., ~5% (3%–10%), suggest that these enhanced particles are solid particles, rather than spherical, liquid $H_2SO_4$ particles (PDR ~0%) or cirrus ice particles (PDR > 25%–30%). A recent laboratory experiment by Wagner et al. (2020a) showed PDR values of ~9.5% for solid $NH_4NO_3$ particles at 488 nm. (Also, Wagner et al. (2020b) showed electron microscope images of solid $NH_4NO_3$ particles, which are "of rather compact shape with aspect ratios predominantly in the range from 0.80 to 1.25.") Thus, the values obtained with our lidars in Japan might be consistent with those of solid $NH_4NO_3$ particles suggested by Höpfner et al. (2019). (Note that Sakai et al. (2010) investigated PDR values of other particle types at 532 nm in laboratory experiments; among these particles, sub-micrometre sea-salt and ammonium sulphate crystals (e.g., Plate 9 (pages 237–239) of Pruppacher and Klett, 1997) were found to have PDR values of ~8% and ~4%, respectively.) Small non-zero PDR values can occur if enhanced liquid $H_2SO_4$ particles and fresh ash particles from volcanic eruptions are mixed, although satellite data indicate this is less plausible (Sect. 3.3). However, it should be noted that the lidar BSR and PDR measurements cannot exclude the possibility of co-existence of other types of solid aerosol particles such as mineral dust (e.g., modelling work by Lau et al., 2018; in situ measurements by Vernier et al., 2018), black carbon (e.g., modelling work by Gu et al., 2016), and some types of carbonaceous aerosols (e.g., modelling works by Gu et al., 2016; Lau et al., 2018; Fairlie et al., 2020) which are solid.

Lower-stratospheric aerosol enhancement over Japan was observed mainly during August–September, and seldom in July. This may be partly explained by the seasonality of the concentration of solid $NH_4NO_3$ particles in the ASM anticyclone (Höpfner et al., 2019), peaking in August with significant year-to-year variations. Furthermore, June and July are in the rainy season for most of Japan, in association with the "Baiu" frontal system (e.g., Ninomiya and Shibagaki, 2007). In July 2018, severe rainfall and flood events occurred early in the month (Shimpo et al., 2019), after which many parts of Japan experienced high surface temperatures with cumulonimbus clouds in several areas. Typhoons, synoptic low systems, and frontal systems affected Japan through to the end of September 2018, with these rainfall and thick-cloud events preventing the lidars from sensing the lower stratosphere, causing many of the missed-data slots in Fig. 1.

In summary, part of the ATAL in the ASM anticyclone airmass is transported eastward and passes over Japan in the UTLS. Lidars in Japan can observe the lower stratospheric portion of these aerosol particles if conditions permit, with summer-time active convection and various weather systems often preventing their sensing of the lower stratosphere. Volcanic eruptions and extensive wildfires may complicate the detection of particles of ATAL origin over Japan. The upper tropospheric portion of these particles is either depleted by tropospheric processes (convection and wet scavenging) or obscured by much stronger cirrus-cloud signals. Despite the limited sampling, the lidar detection of ATAL particles verifies eastward UTLS transport associated with sub-seasonal-scale dynamics of the ASM anticyclone, a process observed by satellite instruments and predicted by models. The spatial extent and chemical and aerosol content of this transport process are the main focus of an upcoming airborne field campaign, the Asian summer monsoon Chemical and Climate Impact Project (ACCLIP; Pan et al., 2019), which is scheduled to take place over the western Pacific during July–August 2021.

## Appendix A

The time–height distribution of TDR at Tsukuba is shown in Figure A1, complementing Fig. 1b (PDR distribution).

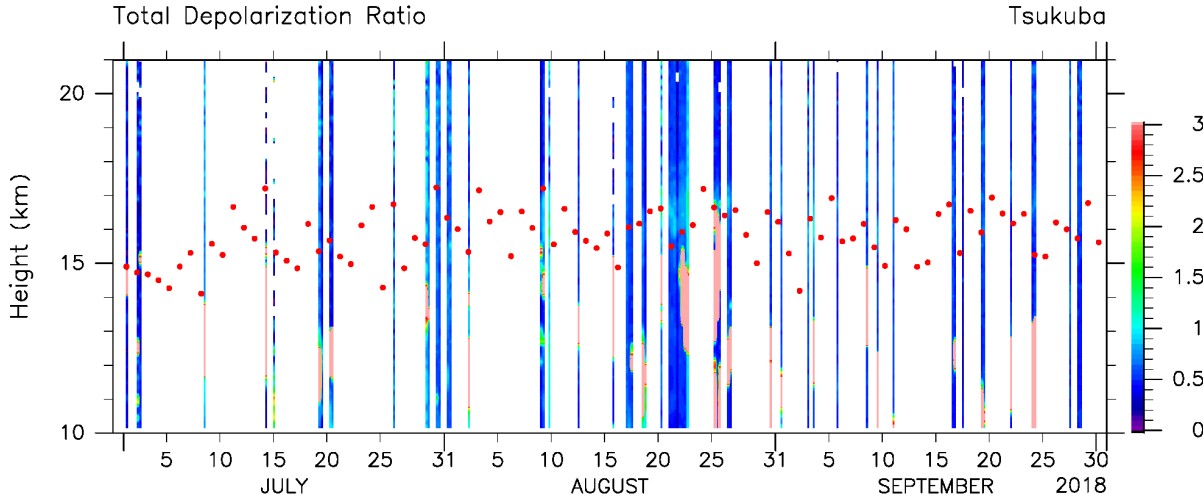

**Figure A1** As for Fig. 1, but for the total depolarization ratio (TDR, %).


### *Data availability*

Lidar data analysed in this study can be downloaded from the following websites: https://mri-2.mri-jma.go.jp/owncloud/s/GrNGNiGKzq8tjqH for Tsukuba; and https://www.cis.fukuoka-

u.ac.jp/~ksiraisi/LidarDataArchive/Fukuoka_2018summer.zip for Fukuoka. ERA5 and CAMS data can be downloaded from the Copernicus website, with the former from https://cds.climate.copernicus.eu and the latter from https://ads.atmosphere.copernicus.eu. MLS Version 4.2 Level 2 data can be downloaded from https://acdisc.gesdisc.eosdis.nasa.gov/data/Aura_MLS_Level2/. OMPS LP Level 2 Version 1.5 data can be downloaded from https://snpp-omps.gesdisc.eosdis.nasa.gov/data. CALIOP data can be downloaded from https://asdc.larc.nasa.gov/search.


### *Author contributions*

MF, MS, and LLP designed the study. T Sakai, TN, and KS operated the lidar systems, and MF, T Sakai, and KS analysed lidar data and drafted the manuscript. YI calculated trajectories. MF and HX analysed CAMS data. SK and MF analysed MLS data. SK analysed OMPS LP data, while T Shibata analysed CALIOP data. All authors contributed to the interpretation, and

reviewed and edited the manuscript.

*Competing interests*

The authors declare that they have no conflict of interest.

*Acknowledgements*

This study was financially supported by the research grant for Mission Research on Sustainable Humanosphere from Research Institute for Sustainable Humanosphere (RISH), Kyoto University, Japan, for the fiscal years 2019–2020. We thank the undergraduate students at Faculty of Science, Fukuoka University who operated the lidar system at Fukuoka. The GFD-DENNOU library was used for producing Figures 1–8 and A1. We thank Nawo Eguchi and Suginori Iwasaki for valuable

comments on draft manuscript. We also thank two anonymous reviewers and Michelle Santee for valuable comments and suggestions.

*Financial support*

This study was financially supported by the research grant for Mission Research on Sustainable Humanosphere from Research

Institute for Sustainable Humanosphere (RISH), Kyoto University, Japan for the fiscal years 2019–2020.

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
