# Peer review of "Lower-stratospheric aerosol measurements in eastward shedding vortices over Japan from the Asian summer monsoon anticyclone during the summer of 2018"

_Atmospheric Chemistry and Physics, 2020_

## Referee Comment (RC1) · Anonymous Referee #1 · 16 Nov 2020

In this manuscript, Fujiwara and co-authors present observational data from two lidars in Japan indicating the presence of aerosol particles from the Asian Tropopause Aerosol Layer (ATAL) over the stations just above the tropopause in August and September 2018. The origin of these particles from inside the Asian monsoon anticyclone is indicated by trajectory analyses as well as CAMS reanalysis data of CO and MLS satellite observations of water vapor. Further, the authors exclude the influence of volcanic eruptions and forest fires by inspecting the global condition of aerosols in the UTLS through space borne limb-scatter (OMPS LP) and lidar (CALIOP) observations.

A central point of this work is the depolarization of the observed particles of around 5%. This indicates that at least a part of the aerosols was non-spherical (i.e. not liquid) but solid, however, not as strongly depolarizing as cirrus particles.

In general, the manuscript is well written and logically organised with clear figures. It presents a novel dataset on aerosols from the ATAL embedded in the interpretation of the general atmospheric situation. To my knowledge it is also the first one of an eastward shedding event over Japan and explicitly including the particle depolarization. Therefore, I recommend publication in ACP after taking into account the comments below.

**Specific comments**

L97-L110, Lidar error estimation:

In L98 the BSR uncertainties are stated as 2-3%, but in L106, additional BSR errors are discussed. I would suggest to clearly state first all error terms for the BSR (random and possible systematic ones) and then include those in the discussion on PDR.

L135, 'in this data product, clouds and Polar Stratospheric Clouds (PSCs) have been removed':

Please add the information, how cirrus clouds have been removed.

L145:

Please discuss also the event around Aug, 9th, over Tsukuba since there has been a clear eastward shedding as can be seen in Figs. 7 and 8. It is not clear to me if the particles were above or below the tropopause since there is quite a strong change in tropopause height visible by the red dots in Fig. 1.

L181, 'PDR values of 1%-3%':

The difference between the PDR values between Tsukuba, showing clearly enhanced signals, and Fukuoka is tentatively explained by the different measurement periods.

**ACPD**
In case of Tsukuba, quasi no event has been shown with values of PDR less than 2 (see Fig. 2), while there are many above Fukuoka. Please discuss whether this might hint to some unidentified bias in one of the instruments. It would e.g. be informative to present some observations before June or after September where both instruments show consistently low/high values of PDR.

L196, '3.2 Trajectories and airmasses':

From the trajectories shown, it is not clear if they reach altitudes below the tropopause. Could you provide any discussion on this point?

L240:

I would be interested if the CAMS CO data could be supported by MLS measurements of CO. This should be easy by providing a figure similar to Fig. 8 but for MLS CO.

L283:

Is any direct comparison/match of the ground based lidars with the CALIOP lidar possible during the relevant time period?

L317:

For ATAL studies, the applied CALIOP filter on cirrus clouds has been a depolarization ratio threshold of 5% (e.g. Vernier et al., 2015): (1) why has a different limit been applied for the present ground based observations? (2) could you discuss which effect the finding of this work indicating 5% depolarization and more for ATAL particles would have on the CALIOP data analysis?

L327, 'with an average BSR value of 1.05 being a systematic feature':

Is there any information on the depolarization available from the OHP-lidar?

L337:

One may add the information from Fig. 2 in Wagner et al., 2020a, that the depolariza-

**ACPD**
tion ratio obtained in the laboratory for solid ammonium nitrate particles was around 9%. Further, in Wagner et al., 2020b, from electron microscope images of ammonium nitrate particles Fig. 2c reveals 'that the crystalline AN particles are of rather compact shape with aspect ratios predominantly in the range from 0.80 to 1.25.'

**Technical corrections**

Figure 4:

The CO isolines for different months cannot be distinguished easily. Perhaps use different line styles.

L223:

Please add in this sentence the information 'from the CAMS reanalysis data'.

L224, 'A potential temperature of 400 K corresponds to altitudes of 17.1 km at Tsukuba and 17.3 km at Fukuoka, on average, during July–September 2018':

This information should be provided before, e.g. where the trajectories are discussed.

L286, 'not have reached':

'have not reached'

L298, '. Rectangular':

'. The rectangular'

L341, 'France, any enhancement':

'France, no enhancement'

**References**

Vernier, J.-P., Fairlie, T. D., Natarajan, M., Wienhold, F. G., Bian, J., Martinsson, B. G., Crumeyrolle, S., Thomason, L. W., and Bedka, K. M.: Increase in upper tropospheric and lower stratospheric aerosol levels and its potential connection with

**ACPD**
Asian pollution, Journal of geophysical research. Atmospheres JGR, 120, 1608–1619, https://doi.org/10.1002/2014JD022372, 2015.

Wagner, R., Bertozzi, B., Höpfner, M., Höhler, K., Möhler, O., Saathoff, H., and Leisner, T.: Solid Ammonium Nitrate Aerosols as Efficient Ice Nucleating Particles at Cirrus Temperatures, J. Geophys. Res., 125, e2019JD032248, https://doi.org/10.1029/2019JD032248, 2020a.

Wagner, R., Testa, B., Höpfner, M., Kiselev, A., Möhler, O., Saathoff, H., Ungermann, J., and Leisner, T.: High-resolution optical constants of crystalline ammonium nitrate for infrared remote sensing of the Asian Tropopause Aerosol Layer, Atmos. Meas. Tech. Discuss., https://doi.org/10.5194/amt-2020-262, 2020b.

---

## Referee Comment (RC2) · Michelle Santee (Referee) · 16 Nov 2020

**Review of "Lower-stratospheric aerosol measurements in eastward shedding vortices over Japan from the Asian summer monsoon anticyclone during the summer of 2018" by Fujiwara et al.**

Ground-based lidar measurements obtained at two stations in Japan during July to September 2018 are analyzed to look for signatures of the transport of aerosols from the Asian summer monsoon (ASM). Particle enhancements were observed in the lower stratosphere in August and September; back trajectories and satellite and reanalysis data are used to show that those air masses originated within the ASM anticyclone and were not influenced by volcanic or biomass burning emissions, and thus they likely reflect extension of the Asian Tropopause Aerosol Layer (ATAL) associated with eastward eddy shedding events. The analysis presented is sound, the manuscript is well written and well organized, and the topic is timely and of interest to the ACP readership. Although I do have some substantive issues that I would like to see considered before the paper is accepted for publication, most of my comments are minor wording suggestions.

**Specific comments and questions (major substantive issues and minor points of clarification, wording suggestions, and grammar / typo corrections are listed together for each Section in sequential order through the manuscript):**

**Introduction**
- L41-42: It would be better to add "e.g." in front of the list of references for aerosols and water vapor, as is done for trace gases.
- L53: was believed --> is believed
- L57-59: The way this sentence is constructed – first talking about the behavior observed during a specific week in August 1997 and then stating that the peak in $NH_4NO_3$ occurs around August – may be slightly confusing for readers, especially those who are not familiar with Höpfner et al.'s paper and the particular satellite data they analyzed. I can understand that the authors do not wish to add extraneous detail to the Introduction, but I think it would be better to break this sentence in two and make it more clear that the findings reported by Höpfner et al. were based on different satellite data sets. As it is now, the week of 8–16 August 1997 appears to hold some special significance, rather than just being when CRISTA data were taken.
- L70: data --> the data

**Section 2**
- L86: made --> done
- L97: The uncertainties of lidar data are discussed here, which are applied to the both systems --> The uncertainties of lidar data discussed here are applicable to both systems
- L121: Since "PV" is used below, "(PV)" should be added here after "potential vorticity".
- L122 and L126: on to --> onto
- L124: Delete the comma after "but".
- L126-127: PV … are --> PV … is
- L136-137: Young and Vaughan (2009) does not appear in the reference list.

**Section 3**
- L157: mixture --> a mixture

- Figure 1: The color bar, particularly for the PDR panel, could be improved. The pink color denoting the highest values is somewhat difficult to distinguish from the light purple used at the bottom of the PDR range; this would not really be a problem if more were filled in, but it complicates interpretation of such a sparse plot.
- L179: Delete the comma after "2018".
- L191: For clarity, it might be good to repeat verbatim the description in the Figure 1 caption: "the daily (first) lapse-rate tropopause".
- L192: with --> at
- L199: at a 400 K --> at 400 K
- L200: of boundaries --> of the boundary
- L201: The value for the CO concentration (65 ppbv) selected to identify the ASM anticyclone boundary seems reasonable, but nevertheless it would be appropriate to cite a reference to justify this choice.
- L210: Although it's implicit, rather than stating "during July–September 2018", it would be better to say "on all days during July–September 2018 on which measurements were made".
- Figures 4 and 5:
  - The vast majority of back trajectories launched from Tsukuba indicate that the air parcels had been transported from altitudes above about 10–12 km over the preceding 10 days (and even slightly higher than that for trajectories run back from Fukuoka). It therefore seemed a bit odd to me that the color bar extends down to Z = 4 km. I had to look closely to spot the single trajectory that appears to originate in the middle of the North Pacific at that altitude (Figure 4, bottom panel). Some explanation for this apparently anomalous parcel should be given.
  - The greens in these color bars are impossible to distinguish, so if the authors feel that the trajectory geopotential height information is important, then they need to use a different color palette. Also, the color bar increments (0.8 km) are awkward.
  - Perhaps both of the above issues could be addressed by reformulating the plots. The existence of the outlier parcel could simply be mentioned in the text and not shown, allowing the geopotential height range to be decreased so fewer colors are needed.
- L230-232: I don't find the discussion of PV very illuminating. First, for clarity it would be better to say "with lower values inside than outside the ASM anticyclone at the same latitude (e.g., 30°N)". More importantly, I'm puzzled by the lack of a clear signature of anticyclonic flow in the PV field. Previous studies have diagnosed eddy shedding through examination of PV maps. In particular, Garny and Randel [JGR, 2013] (a paper that I am surprised to see is not cited in this manuscript) reported an episode of eastward eddy shedding in June 2006 not unlike the one depicted here, and they showed that the contours of CO (from MLS) and PV (from MERRA) follow very similar patterns (although they focused on a lower theta level, 360 K). Perhaps the authors should explore using different PV contours or applying some smoothing to the PV fields to see if a clearer signal emerges. In any case, more discussion of the behavior of PV associated with this event is warranted.
- L236: It seems to me that, in addition to Figure 7, the results of Figure 6 also suggest that the 60-ppbv CO contour is indicative of eastward eddy shedding vortices.
- L237-238: It is difficult for the reader to judge the timing of these events from Figure 7, especially given that some y-axis tick marks appear to be absent. For example, for the first episode listed, the 60-ppbv CO contour seems to be present at the longitudes of the lidar sites a

couple of days before 5 August. Are the 20-25 and 27-31 August events really separable? It might be helpful to guide the eye by overlaying colored horizontal lines to mark each event's start and end dates or perhaps pale (transparent) colored bands spanning the intervals.

- L223-240: Since previous studies have used MLS CO to look for evidence of eddy shedding (e.g., Garny and Randel [2013], Honomichl and Pan [2020]), and since MLS water vapor is shown in Figure 8, I am curious why only CO from the CAMS reanalysis – and not from MLS – is used here. Is it because the necessary longitudinal and temporal binning would smear out finer-scale structures too much? Clearly, despite such smoothing, the MLS $H_2O$ data provide valuable information. On the flip side, the authors should probably offer an explanation of why water vapor from MLS was used but not that from CAMS.
- L268-269: types of particle and gas --> types of particles and gases
- L286: but not have --> but had not
- L287: I think it would be appropriate to add "at least not in a monthly mean view" at the end of this sentence (mirroring the zonal mean caveat in L282).
- L288: unlikely due to --> unlikely to be due to

**Section 4**
- L311: measurements at --> measurements made at; also, delete the comma after "2018"
- L314: the BSR --> BSR
- L315-316: The authors might want to specifically note that none of the 11 nights on which data were taken at Fukuoka fell during the intervals of strong enhancement observed at Tsukuba.
- L318: are of a few --> are a few
- L319: originate in the ASM anticyclone in association with eastward shedding vortices --> originate in the ASM anticyclone and are transported over these sites in association with eastward shedding vortices
- L320: eruptions and extensive --> eruptions or extensive
- L325-330: I'm not convinced that this is the best place for the discussion of the OHP measurements. I think it is generally inappropriate to introduce new aspects in the "Summary and Conclusions" section. In fact, these lines might belong in the Introduction.
- L332: Add a comma after "(3%–10%)".
- L337: Is it necessary to repeat "PDR" 3 times in this line (i.e., is it needed after "8%" and "4%")?
- L340-342: It seems to me that here again the Moana Loa and OHP information in these lines is out of place. Since these data add to the evidence that the signals observed at Tsukuba and Fukuoka must have arisen from the ATAL, this discussion could be moved to Section 3.3, which could then be renamed to reflect its exploration of other potential causes of the observed enhancements and not just focus on "Satellite aerosol data".
- L341: any enhancement --> no enhancement
- L360: Delete the comma after "extent"; also, is --> are

**References**
- L440: The paper by Hanumanthu et al. has now been accepted for publication in ACP.
- L473: The reference for the MLS Data Quality Document (Livesey et al.) is missing the year.

---

## Referee Comment (RC3) · Anonymous Referee #3 · 19 Nov 2020

Comments on Manuscript No. ACP-2020-980

In the last decade the Asian Tropopause Aerosol Layer (ATAL) becomes in the focus of attention. This study shows that the transport of aerosol out of the ATAL by eastward shedding vortices were measured over Japan by two lidar systems during summer 2018. Several eddy shedding events were observed and backward trajectory calculations indicate that eddies including air masses with enhanced aerosol particles originate in the Asian monsoon anticyclone. The analysis of satellite observations and meteorological reanalysis confirm the eddy events and further show that the consid-

ered time period was free of the impact of volcanic eruptions and high forest fires.

This is a very interesting study, which merits its publication in ACP. The scientific content, the quality of the study and its presentation is good. Therefore, I suggest only some minor revisions before publication by ACP.

1) P2/L51: 'The enhanced aerosol particle signature in the ASM anticyclone at 14–18 km altitude is known as the Asian Tropopause Aerosol Layer (ATAL), which was believed to consist of carbonaceous and sulphate materials, mineral dust, and nitrate particles (Vernier et al., 2015, 2018; Brunamonti et al., 2018; Bossolasco et al., 2020; Hanumanthu et al., 2020).'

From this statement it is not clear if the knowledge about the chemical composition of the the ATAL particles is based on in situ measurements, remotes sensing observations or model simulations. The discussion about the chemical composition of the ATAL should be much more improved and clarified. I recommend to add a short summery about the current knowledge of ATAL particle characteristics (e.g. chemical composition, particle size distribution, particle form, possible sources etc.). This would help to better bring the results of the lidar measurements presented in this study into the context of previous publications.

2) P4/L114: ERA5 is a very new product from ECMWF, therefore I think it is worth to add a few references demonstrating the quality of ERA5 compared to the former ERA-Interim reanalysis.

3) P4/L122: Please add a statement like this: 'CO and the ATAL have not necessarily the same emission sources, however CO is a good chemical tracer to indicated the location of the Asian monsoon anticyclone.'

4) P5/L152: Is it possible that cirrus signal overlays the aerosol signal, so that cirrus and aerosol can coexist simultaneously? Or can you exclude this with your method?

5) P9/Fig.3: You could add a BSR profile from pre- or post-monsoon to show the difference. The difference can be use to better highlight the signal in BSR from the ATAL.

6) P10/L204: '... whereas those without enhanced aerosol particles tend to originate from edge regions surrounding the anticyclone.' and from the extratropical lower stratosphere. Right?

Why do you use only ten-days backward trajectories? What about 15- or 20-day backward calculations? In somewhat longer trajectories, the difference between air masses from the core the anticyclone or from the edge (or outside from the anticyclone) should be more pronounced.

7) P11/Fig.4: Is it possible to adjust the color bar more to the Z range of the trajectories to better highlight the gradients along the trajectories. The bluish colors are only used for one trajectory over the Pacific in Fig. 4b. It looks like that this trajectory is influenced by a tropical cyclone. If that is true that could be mentioned as a side remark.

8) P13/L230 : 'PV can be regarded as a dynamical tracer, with lower values in the ASM anticyclone along the same latitudes (e.g., 30°N), although background positive gradients in latitude and its noisier nature give more complicated features.'

PV can be very useful to see the edge of the Asian monsoon anticyclone at around 380K (e.g. Ploeger et al., 2015), above around 400K as shown in Fig. 6, the PV is not so useful. Instead you could try to use the (Montgomery) stream function or the geopotential height.

Ploeger, F., Gottschling, C., Grießbach, S., Grooß, J.-U., Günther, G., Konopka, P., Müller, R., Riese, M., Stroh, F., Tao, M., Ungermann, J., Vogel, B., and von Hobe, M.: A potential vorticity-based determination of the transport barrier in the Asian summer monsoon anticyclone, Atmos. Chem. Phys., 15, 13 145–13 159, https://doi.org/doi:10.5194/acp-15-13145-2015, 2015.

9) P16/Fig.8: Why do you show $H_2O$ from MLS and not CO from MLS? CO would be a better chemical tracer for transport as $H_2O$ which is in addition affected by micro-

Interactive
comment

physics. You could also use MLS O3 which should be anticorrelated to CO (low O3 in the anticyclone and high O3 in the lower stratosphere).

10) P20/L331: 'The PDR values obtained at Tsukuba, i.e., ~5% (3%–10%) suggest that these enhanced particles are solid particles, rather than spherical, liquid H2SO4 particles (PDR ~0%) or cirrus ice particles (PDR > 25%–30%). The observed values may be consistent with those of solid NH4NO3 particles recently suggested by Höpfner et al. (2019).'

Using the particle depolarization ratio, the study shows that the aerosol particles are most likely solid and it is concluded that the aerosol particles possibly contain NH4NO3. In the literature it is discussed that also carbonaceous aerosols, dust, nitrate-containing aerosol, black carbon and organic carbon could contribute to the chemical composition of the ATAL. Can you exclude with your measurements such types of aerosol particles? Please clarify this point.

Minor comments:

1) P3/L84: (senkrecht in German) –> (= "senkrecht" in German) ?

2) P5/L37: remove large white spaces

3) P13/L223: 'Horizontal distributions of CO and PV' add 'from CAMS'

4) P13/L233: 'are shown in Figure 7' -> 'are shown as Hovmöller diagrams in Fig. 7'

5) Fig.7/8: You could say that the Figures are 'Hovmöller diagrams'
* * *

---

## Author Comment (AC1) · 16 Dec 2020

**Response to comments by Anonymous Referee #1**

Thank you very much for your review.

In this manuscript, Fujiwara and co-authors present observational data from two lidars in Japan indicating the presence of aerosol particles from the Asian Tropopause Aerosol Layer (ATAL) over the stations just above the tropopause in August and September 2018. The origin of these particles from inside the Asian monsoon anticyclone is indicated by trajectory analyses as well as CAMS reanalysis data of CO and MLS satellite observations of water vapor. Further, the authors exclude the influence of volcanic eruptions and forest fires by inspecting the global condition of aerosols in the UTLS through space borne limb-scatter (OMPS LP) and lidar (CALIOP) observations.

A central point of this work is the depolarization of the observed particles of around 5%. This indicates that at least a part of the aerosols was non-spherical (i.e. not liquid) but solid, however, not as strongly depolarizing as cirrus particles.

In general, the manuscript is well written and logically organised with clear figures. It presents a novel dataset on aerosols from the ATAL embedded in the interpretation of the general atmospheric situation. To my knowledge it is also the first one of an eastward shedding event over Japan and explicitly including the particle depolarization. Therefore, I recommend publication in ACP after taking into account the comments below.

Thank you very much for your evaluation.

Specific comments

L97-L110, Lidar error estimation:
In L98 the BSR uncertainties are stated as 2-3%, but in L106, additional BSR errors are discussed. I would suggest to clearly state first all error terms for the BSR (random and possible systematic ones) and then include those in the discussion on PDR.

This sentence will be revised as follows:
"The BSR uncertainties were estimated as follows. The random component was estimated from the photon counts of the backscatter signals at 532 nm after temporal and vertical averaging by assuming Poisson statistics. Other sources of BSR uncertainties (biases) were estimated by assuming the uncertainty of the normalization value of BSR with $8.5\times10^{-3}$ (Russell et al., 1979, 1982) and that of the extinction-to-backscatter ratio with 30 sr (Jäger and Hofmann, 1991; Jäger et al., 1995). The total uncertainty of BSR were then estimated to be 2–3 % typically around the tropopause."

References:

Jäger, H., and Hofmann, D.: Midlatitude lidar backscatter to mass, area, and extinction conversion model based on in situ aerosol measurements from 1980 to 1987, Appl. Opt., 30(1), 127, https://doi.org/10.1364/ao.30.000127, 1991.

Jäger, H., Deshler, T., and Hofmann, D. J.: Midlatitude lidar backscatter conversions based on balloonborne aerosol measurements, Geophys. Res. Lett., 22(13), 1729–1732, https://doi.org/10.1029/95GL01521, 1995.

Russell, P. B., Swissler, T. J., and McCormick, M. P.: Methodology for error analysis and simulation of lidar aerosol measurements, Appl. Opt., 18(22), 3783, https://doi.org/10.1364/ao.18.003783, 1979.

Russell, P. B., Morley, B. M., Livingston, J. M., Grams, G. W., and Patterson, E. M.: Orbiting lidar simulations 1: Aerosol and cloud measurements by an independent-wavelength technique, Appl. Opt. 21(9), 1541, https://doi.org/10.1364/ao.21.001541, 1982.

L135, 'in this data product, clouds and Polar Stratospheric Clouds (PSCs) have been removed':
Please add the information, how cirrus clouds have been removed.

The algorithms to classify the CALIOP backscatter signals into various types of clouds and aerosols, presented by Young and Vaughan (2009), are quite complicated, but in short, they are based on the particulate extinction-to-backscatter (lidar) ratio and the multiple-scattering factor profile. At the end of the sentence, we will add "based on the information of particulate extinction-to-backscatter (lidar) ratio and the multiple-scattering factor profile". Also, we will add more recent publication by Kim et al. (2018).

Reference:
Kim, M.-H., Omar, A. H., Tackett, J. L., Vaughan, M. A., Winker, D. M., Trepte, C. R., Hu, Y., Liu, Z., Poole, L. R., Pitts, M. C., Kar, J., and Magill, B. E.: The CALIPSO version 4 automated aerosol classification and lidar ratio selection algorithm, Atmos. Meas. Tech., 11, 6107–6135, https://doi.org/10.5194/amt-11-6107-2018, 2018.

L145:
Please discuss also the event around Aug, 9th, over Tsukuba since there has been a clear eastward shedding as can be seen in Figs. 7 and 8. It is not clear to me if the particles were above or below the tropopause since there is quite a strong change in tropopause height visible by the red dots in Fig. 1.

We will add the following sentence just after describing the 18–26 August case:
"We also observe another strong event around 9 August at 15–17 km, although missing observations before and after this date prevent from characterizing the temporal scale of the event; furthermore, the tropopause height was highly variable around this date and was located at 17 km on that date, situating the aerosol enhanced layer temporarily in the troposphere."
We will also add this information to Section 3.2 in the discussion for Figure 7 as ". . . in the fact that only the 20–25

August event was relatively well observed, with the 5–15 August event being captured only on 9 August."

L181, 'PDR values of 1%–3%':

The difference between the PDR values between Tsukuba, showing clearly enhanced signals, and Fukuoka is tentatively explained by the different measurement periods. In case of Tsukuba, quasi no event has been shown with values of PDR less than 2 (see Fig. 2), while there are many above Fukuoka. Please discuss whether this might hint to some unidentified bias in one of the instruments. It would e.g. be informative to present some observations before June or after September where both instruments show consistently low/high values of PDR.

The reason why PDR values less than 2% were rare in the plot for Tsukuba (Fig. 2) is that we do not plot the data with "relative" uncertainty of PDR larger than 30%; this treatment resulted in removing data points with BSR values lower than ~1.05 where background spherical sulphate particles (with PDR values of <2%) were presumably predominant. We will note this treatment for Tsukuba data in the revised manuscript. Figures R1-1 and R1-2 below show the lidar profiles at Tsukuba on some days in May 2018 and in October 2018, respectively. These figures show that BSR in these months did not show enhancements of >1.1 like those found in August and September (Figs. 2c, e). However, we note that TDR slightly increased below 20 km, suggesting a possibility of presence of minute amount of non-spherical particles. The origins of the particles are unknown and a subject of our future study.

Figure R1-3 below shows the lidar profiles at Fukuoka on some days in May-June 2018 and in October-November 2018. Again, we did not detect depolarization enhanced layers (with depolarization ratio higher than 2%) in these months.

Tsukuba 2018/05/02 00:01:51-04:18:47LST

Tsukuba 2018/05/04 19:03:46-00:59:49LST

Tsukuba 2018/05/10 19:04:06-00:59:07LST

Tsukuba 2018/05/13 00:04:52-00:59:41LST

Tsukuba 2018/05/14 19:06:45-00:58:32LST

Tsukuba 2018/05/15 19:07:38-00:59:50LST

Tsukuba 2018/05/18 00:02:54-00:58:03LST

Tsukuba 2018/05/19 19:06:38-00:58:42LST

Tsukuba 2018/05/22 19:08:30-00:58:06LST

Figure R1-1. Lidar profiles taken at Tsukuba on May 2, 4, 10, 13, 14, 15, 18, 19, 22, 24, and 28, 2018. The horizontal dashed line in each panel indicates the location of the first lapse rate tropopause. It is noted that for "TDR*10" = 0.05 means TDR = 0.05/10 = 0.005, i.e., 0.5%TDR. This is close to the depolarization ratio value, 0.366% for air molecules.

Tsukuba 2018/10/02 19:34:54-00:59:27LST
Tsukuba 2018/10/06 19:01:21-00:58:31LST
Tsukuba 2018/10/08 19:19:53-00:59:51LST

Tsukuba 2018/10/09 19:01:14-00:59:24LST
Tsukuba 2018/10/18 19:00:24-00:25:19LST
Tsukuba 2018/10/21 19:01:46-00:57:00LST

Tsukuba 2018/10/24 19:00:27-00:59:22LST
Tsukuba 2018/10/25 19:00:27-00:59:42LST
Tsukuba 2018/10/28 19:03:08-00:59:09LST

[Figure]

Figure R1-2. As for Figure R1-1, but for October 2, 6, 8, 9, 18, 21, 24, 25, 28, 29, and 30, 2018.

[Figure]

Figure R1-3. Lidar profiles taken at Fukuoka on the days in May-June 2018 and in October-November 2018 when the lidar was operated.

L196, '3.2 Trajectories and airmasses':

From the trajectories shown, it is not clear if they reach altitudes below the tropopause. Could you provide any discussion on this point?

Following the comments by other reviewers, the colored geopotential height range of Figures 4 and 5 will be changed (narrowed); please see the revised version of these figures at the end of this response letter.

We will revise the sentence,

"They also indicate that airmasses with enhanced aerosol particles at this height tend to originate in regions within the ASM anticyclone, whereas those without enhanced aerosol particles tend to originate from edge regions surrounding the anticyclone."

as:

"They also indicate that airmasses with enhanced aerosol particles at this height tend to originate in regions within the ASM anticyclone at the altitudes 16.5–18 km, i.e., around or just below the tropopause, whereas those without enhanced aerosol particles tend to originate from edge regions surrounding the anticyclone." (i.e., the underlined part will be added.)

L240:

I would be interested if the CAMS CO data could be supported by MLS measurements of CO. This should be easy by providing a figure similar to Fig. 8 but for MLS CO.

Thank you for this suggestion. Please see Figure R1-4 below. We see that CAMS CO data are roughly ~10 ppbv greater than MLS CO over Japan during August–September 2018, but also that eastward extension signals coming over Japan agree fairly well qualitatively within the differences in spatio-temporal sampling of the two data sets. We will add this note to Section 2.2.

[Figure]

Figure R1-4. (Left) Same as Figure 7 (the revised version following Referee #2's suggestion: Coloured line segments have been added for the periods when CO concentration was ≥~60 ppbv along the longitude of Tsukuba (i.e., 3–15, 20–24, and 28–31 August, and 3–8, 14–17, and 28–29 September)). (Right) Same as left but for MLS CO data. Data for the 30°N–40°N region have been aggregated into 3-day and 8°-longitude bins, each constituting about 10 individual data points. The contours for 45 ppbv and 55 ppbv as well as 65 ppbv are added as dotted lines.

L283:

Is any direct comparison/match of the ground based lidars with the CALIOP lidar possible during the relevant time period?

We have looked at the CALIOP lidar data on 21 August 2018 when the lidar at Tsukuba (36.1°N, 140.1°E) observed the strong signal and when the CALIPSO flew over the region relatively close to Tsukuba: https://www-calipso.larc.nasa.gov/products/lidar/browse_images/show_v4_detail.php?s=production&v=V4-10&browse_date=2018-08-21&orbit_time=03-36-39&page=3&granule_name=CAL_LID_L1-Standard-V4-10.2018-08-21T03-36-39ZD.hdf

As you see, we do not observe the corresponding signals around the orbital track (31.71°N, 139.38°E) to (37.78°N, 137.66°E) around 15.5–18 km in CALIOP data. Young and Vaughan (2009) noted for the CALIOP measurements, "The usual method of increasing the SNR (signal-to-noise ratio) is to average many profiles. However, along-track inhomogeneities in the atmospheric features, combined with the high speed (typically 7.5 km s$^{-1}$) of the satellite across these features and the relatively low firing rate of the laser ($\sim$20 s$^{-1}$), lead to situations where it is simply not possible to acquire a sufficient number of profiles before the subsatellite atmosphere changes significantly." Note also that the ground-based lidar data from Tsukuba (Fukuoka) have been averaged for 3 (4) hours in this study. We think that appropriate quality control (e.g., taking only high-quality nighttime measurements) and spatio-temporal averaging for CALIOP data might give us ATAL eastward extension signals over Japan in a Hovmöller diagram like Figures 7 and 8, but this would be a potential future work.

L317:

For ATAL studies, the applied CALIOP filter on cirrus clouds has been a depolarization ratio threshold of 5% (e.g. Vernier et al., 2015): (1) why has a different limit been applied for the present ground based observations? (2) could you discuss which effect the finding of this work indicating 5% depolarization and more for ATAL particles would have on the CALIOP data analysis?

Regarding (1), Vernier et al. (2009) masked ice clouds by using their total depolarization ratio (TDR'), defined by the perpendicular to total (particular plus molecular) backscatter signal, and not by using PDR. In that case, TDR' values would be small if BSR values of cirrus are small. Theoretically, for example, TDR' would be less than 5% if BSR values are smaller than 1.2 for cirrus for the case that the PDR value is 35% (see Figure R1-5 below). In addition, averaging the data for a large grid volume (2° longitude × 1° latitude × 200 m height) could reduce BSR and TDR' values if cirrus clouds are partially present in the volume. Thus, we presume that they removed the data even if it contained some cirrus clouds.

Reference:

Vernier, J. P., Pommereau, J. P., Garnier, A., Pelon, J., Larsen, N., Nielsen, J., Christensen, T., Cairo, F., Thomason, L. W., Leblanc, T., and McDermid, I. S.: Tropical stratospheric aerosol layer from CALIPSO lidar observations, J.

Geophys. Res., 114, https://doi.org/10.1029/2009jd011946, 2009.

[Figure]

Figure R1-5. A theoretical total depolarization ratio (TDR') curve as a function of BSR for cirrus cloud particles with the PDR value of 35%. The horizontal dotted red line shows the threshold value of TDR' (5%) used in the papers by Vernier et al. (2009, 2015).

Regarding (2), we do not argue that future ATAL studies using CALIOP data should always use smaller threshold values. The choice of the threshold would depend on the purpose of each study. It is also possible that appropriate threshold numbers differ for different regions, e.g., either over the Tibetan Plateau or over Japan, in part because the chemical composition and crystal shape (if they are solid particles) may differ for different regions due to different life stages of the ATAL particles.

L327, 'with an average BSR value of 1.05 being a systematic feature':
Is there any information on the depolarization available from the OHP-lidar?

Unfortunately, the lidar at OHP has no capability of measuring the depolarization.

L337:
One may add the information from Fig. 2 in Wagner et al., 2020a, that the depolarization ratio obtained in the laboratory for solid ammonium nitrate particles was around 9%. Further, in Wagner et al., 2020b, from electron microscope images of ammonium nitrate particles Fig. 2c reveals 'that the crystalline AN particles are of rather compact shape with aspect ratios predominantly in the range from 0.80 to 1.25.'

Thank you very much for the information of these very important and relevant papers. We will modify the relevant part of the text. The revised, whole paragraph will be as follows:
"The PDR values obtained at Tsukuba, i.e., ~5% (3%–10%) suggest that these enhanced particles are solid particles, rather than spherical, liquid $H_2SO_4$ particles (PDR ~0%) or cirrus ice particles (PDR > 25%–30%). A recent laboratory experiment by Wagner et al. (2020a) showed the PDR values of ~9.5% for solid $NH_4NO_3$ particles at 488 nm. (Also, Wagner et al. (2020b) showed electron microscope images of solid $NH_4NO_3$ particles, which are "of rather compact shape with aspect ratios predominantly in the range from 0.80 to 1.25.") Thus, the values obtained with our lidars in Japan might be consistent

with those of solid NH$_4$NO$_3$ particles suggested by Höpfner et al. (2019). (Note that Sakai et al. (2010) investigated PDR values of other particle types at 532 nm in laboratory experiments; among these particles, sub-micrometre sea-salt and ammonium sulphate crystals (e.g., Plate 9 (pages 237–239) of Pruppacher and Klett, 1997) were found to have PDR values of ~8% PDR and ~4% PDR, respectively.) Small non-zero PDR values can occur if enhanced liquid H$_2$SO$_4$ particles and fresh ash particles from volcanic eruptions are mixed, although satellite data indicate this is less plausible (Sect. 3.3). Lidar measurements at Mauna Loa, Hawaii, indicated no signals from volcanic eruptions during the summer of 2018 (Chouza et al., 2020). Also, at the OHP lidar site in France, no enhancement in the lower stratospheric aerosol abundance was observed during the summer of 2018."

Technical corrections

Figure 4:
The CO isolines for different months cannot be distinguished easily. Perhaps use different line styles.

We will revise this figure as suggested. (Please see the revised version of these figures at the end of this response letter.)

L223:
Please add in this sentence the information 'from the CAMS reanalysis data'.

We will add this information.

L224, 'A potential temperature of 400 K corresponds to altitudes of 17.1 km at Tsukuba and 17.3 km at Fukuoka, on average, during July–September 2018':
This information should be provided before, e.g. where the trajectories are discussed.

This information will be moved to the first paragraph of Section 3.2.

L286, 'not have reached':
'have not reached'

Will be corrected.

L298, '. Rectangular':
'. The rectangular'

Will be corrected.

L341, 'France, any enhancement':
'France, no enhancement'

Will be corrected.

References

Vernier, J.-P., Fairlie, T. D., Natarajan, M., Wienhold, F. G., Bian, J., Martinsson, B. G., Crumeyrolle, S., Thomason, L. W., and Bedka, K. M.: Increase in upper tropospheric and lower stratospheric aerosol levels and its potential connection with Asian pollution, Journal of geophysical research. Atmospheres JGR, 120, 1608–1619, https://doi.org/10.1002/2014JD022372, 2015.

Wagner, R., Bertozzi, B., Höpfner, M., Höhler, K., Möhler, O., Saathoff, H., and Leisner, T.: Solid Ammonium Nitrate Aerosols as Efficient Ice Nucleating Particles at Cirrus Temperatures, J. Geophys. Res., 125, e2019JD032248, https://doi.org/10.1029/2019JD032248, 2020a.

Wagner, R., Testa, B., Höpfner, M., Kiselev, A., Möhler, O., Saathoff, H., Ungermann, J., and Leisner, T.: High-resolution optical constants of crystalline ammonium nitrate for infrared remote sensing of the Asian Tropopause Aerosol Layer, Atmos. Meas. Tech. Discuss., https://doi.org/10.5194/amt-2020-262, 2020b.

The two papers by Wagner et al. will be cited in the revised manuscript. The paper by Vernier et al. has already been cited.

[Figure]

Revised Figure 4: The colour code for geopotential height has been changed (to narrow the range of Z). The CO isolines for different months are expressed with different line styles (i.e., dotted for July, solid for August, and dash-dotted for September).

[Figure]

Revised Figure 5: The colour code for geopotential height has been changed (to narrow the range of Z). The CO isolines for different months are expressed with different line styles (i.e., dotted for July, solid for August, and dash-dotted for September).

---

## Author Comment (AC2) · 16 Dec 2020

**Response to comments by Referee #2 (Dr. Michelle Santee)**

Thank you very much for your review.

**Review of "Lower-stratospheric aerosol measurements in eastward shedding vortices over Japan from the Asian summer monsoon anticyclone during the summer of 2018"**
**by Fujiwara et al.**

Ground-based lidar measurements obtained at two stations in Japan during July to September 2018 are analyzed to look for signatures of the transport of aerosols from the Asian summer monsoon (ASM). Particle enhancements were observed in the lower stratosphere in August and September; back trajectories and satellite and reanalysis data are used to show that those air masses originated within the ASM anticyclone and were not influenced by volcanic or biomass burning emissions, and thus they likely reflect extension of the Asian Tropopause Aerosol Layer (ATAL) associated with eastward eddy shedding events. The analysis presented is sound, the manuscript is well written and well organized, and the topic is timely and of interest to the ACP readership. Although I do have some substantive issues that I would like to see considered before the paper is accepted for publication, most of my comments are minor wording suggestions.

Thank you very much for your evaluation.

**Specific comments and questions (major substantive issues and minor points of clarification, wording suggestions, and grammar / typo corrections are listed together for each Section in sequential order through the manuscript):**

**Introduction**
- L41-42: It would be better to add "e.g." in front of the list of references for aerosols and water vapor, as is done for trace gases.

Will be added.

- L53: was believed --> is believed

Will be changed.

- L57-59: The way this sentence is constructed – first talking about the behavior observed during a specific week in August 1997 and then stating that the peak in $NH_4NO_3$ occurs around August – may be slightly confusing for readers, especially those who are not familiar with Höpfner et al.'s paper and the particular satellite data they analyzed. I can understand that the authors do not wish to add extraneous detail to the Introduction, but I think it would be better to break this sentence in two and make it more clear that the findings reported by Höpfner et al. were based on different satellite data sets. As it is now, the week of 8–16 August 1997 appears to hold some special significance, rather than just being when CRISTA data were taken.

We will revise this sentence as: "Their satellite data analysis using Cryogenic Infrared Spectrometers and Telescopes for the Atmosphere (CRISTA) data indicates enhanced $NH_4NO_3$ signals around the tropopause, both in the ASM region and the western Pacific (including Japan) during 8‑16 August 1997 (with the western Pacific signals suggestive of shedding vortices); also, their analysis of satellite Michelson Interferometer for Passive Atmospheric Sounding (MIPAS) data together with CRISTA data show that the mass of $NH_4NO_3$ in the ASM region at 13–17 km peaks around August."

- L70: data --> the data

Will be corrected.

**Section 2**
- L86: made --> done

Will be changed.

- L97: The uncertainties of lidar data are discussed here, which are applied to the both systems --> The uncertainties of lidar data discussed here are applicable to both systems

Will be changed.

- L121: Since "PV" is used below, "(PV)" should be added here after "potential vorticity".

Will be added.

- L122 and L126: on to --> onto

Will be corrected.

- L124: Delete the comma after "but".

Will be deleted.

- L126-127: PV … are --> PV … is

Will be corrected.

- L136-137: Young and Vaughan (2009) does not appear in the reference list.

Thank you for pointing this out. We will add it.

**Section 3**

- L157: mixture --> a mixture

Will be added.

- Figure 1: The color bar, particularly for the PDR panel, could be improved. The pink color denoting the highest values is somewhat difficult to distinguish from the light purple used at the bottom of the PDR range; this would not really be a problem if more were filled in, but it complicates interpretation of such a sparse plot.

We will change the color bar, by making the pink darker, and by removing light purple from the PDR (and TDR) figure. Around the end of this response letter, we attach the revised version of the figures.

- L179: Delete the comma after "2018".

Will be deleted.

- L191: For clarity, it might be good to repeat verbatim the description in the Figure 1 caption: "the daily (first) lapse-rate tropopause".

Will be added.

- L192: with --> at

Will be corrected.

- L199: at a 400 K --> at 400 K

Will be removed.

- L200: of boundaries --> of the boundary

Will be added.

- L201: The value for the CO concentration (65 ppbv) selected to identify the ASM anticyclone boundary seems reasonable, but nevertheless it would be appropriate to cite a reference to justify this choice.

We will revise this sentence as: "By comparing the results from Santee et al. (2017) with our own analysis, the 65 ppbv contours . . . are chosen . . . "

- L210: Although it's implicit, rather than stating "during July–September 2018", it would be better to say "on all days during July–September 2018 on which measurements were made".

Will be corrected as suggested.

- Figures 4 and 5:
  - The vast majority of back trajectories launched from Tsukuba indicate that the air parcels had been transported from altitudes above about 10–12 km over the preceding 10 days (and even slightly higher than that for trajectories run back from Fukuoka). It therefore seemed a bit odd to me that the color bar extends down to Z = 4 km. I had to look closely to spot the single trajectory that appears to originate in the middle of the North Pacific at that altitude (Figure 4, bottom panel). Some explanation for this apparently anomalous parcel should be given.
  - The greens in these color bars are impossible to distinguish, so if the authors feel that the trajectory geopotential height information is important, then they need to use a different color palette. Also, the color bar increments (0.8 km) are awkward.
  - Perhaps both of the above issues could be addressed by reformulating the plots. The existence of the outlier parcel could simply be mentioned in the text and not shown, allowing the geopotential height range to be decreased so fewer colors are needed.

The coloured geopotential height range of Figures 4 and 5 will be changed (narrowed); please see the revised version of these figures at the end of this response letter.

Regarding the trajectory originating in the middle of the North Pacific at that altitude (Figure 4, bottom panel), we will add the following note in the text: "Note that there is a trajectory that originates in the Pacific south of Japan as low as 4 km (Figure 4, bottom, a small-scale spiral in purple); this is associated with upward transport in the typhoon Soulik."

- L230-232: I don't find the discussion of PV very illuminating. First, for clarity it would be better to say "with lower values inside than outside the ASM anticyclone at the same latitude (e.g., 30°N)". More importantly, I'm puzzled by the lack of a clear signature of anticyclonic flow in the PV field. Previous studies have diagnosed eddy shedding through examination of PV maps. In particular, Garny and Randel [JGR, 2013] (a paper that I am surprised to see is not cited in this manuscript) reported an episode of eastward eddy shedding in June 2006 not unlike the one depicted here, and they showed that the contours of CO (from MLS) and PV (from MERRA) follow very similar patterns (although they focused on a lower theta level, 360 K). Perhaps the authors should explore using different PV contours or applying some smoothing to the PV fields to see if a clearer signal emerges. In any case, more discussion of the behavior of PV associated with this event is warranted.

The paper by Garny and Randel (2013) will be cited in the Introduction.

Following the suggestion by Referee #3 (i.e., at and above 400 K, PV is not very useful to see the boundary of the ASM anticyclone), we will change from PV to Montgomery streamfunction (MSF). Please see the response to Referee #3 for detailed revisions. Around the end of this response letter, we show the revised Figure 6 with MSF.

- L236: It seems to me that, in addition to Figure 7, the results of Figure 6 also suggest that the 60-ppbv CO contour is indicative of eastward eddy shedding vortices.

This sentence is specifically for Figure 7 in which we take a latitudinal average. We will revise this sentence as: "In Fig.7, the 60-ppbv CO contour may be a good indicator of eastward shedding vortices."

- L237-238: It is difficult for the reader to judge the timing of these events from Figure 7, especially given that some y-axis tick marks appear to be absent. For example, for the first episode listed, the 60-ppbv CO contour seems to be present at the longitudes of the lidar sites a couple of days before 5 August. Are the 20-25 and 27-31 August events really separable? It might be helpful to guide the eye by overlaying colored horizontal lines to mark each event's start and end dates or perhaps pale (transparent) colored bands spanning the intervals.

We add coloured line segments for the periods when CO concentration was ≥~60 ppbv along the longitude of Tsukuba. Please see the revised Figure 7 at the end of this response letter. (The same line segments will also be added to the MLS figures.) Furthermore, the text describing these periods will be corrected as: 3–15, 20–24, and 28–31 August, and 3–8, 14–17, and 28–29 September.

- L223-240: Since previous studies have used MLS CO to look for evidence of eddy shedding (e.g., Garny and Randel [2013], Honomichl and Pan [2020]), and since MLS water vapor is shown in Figure 8, I am curious why only CO from the CAMS reanalysis – and not from MLS – is used here. Is it because the necessary longitudinal and temporal binning would smear out finer-scale structures too much? Clearly, despite such smoothing, the MLS $H_2O$ data provide valuable information. On the flip side, the authors should probably offer an explanation of why water vapor from MLS was used but not that from CAMS.

Figures R2-1 and R2-2 below show the comparisons of CAMS data and MLS data for CO and for water vapor. As we can see, for both CO and water vapor, CAMS and MLS show qualitatively and broadly similar eastward extension signals over Japan; however, CAMS CO is greater than MLS CO (e.g., the differences are ~10 ppbv around the longitudes of Japan through August–September 2018), and CAMS water vapor mixing ratios are greater than MLS water vapor (e.g., the differences are roughly ~2 ppmv for the wet signals around the longitudes of Japan in August 2018). In this paper, we primarily use CAMS CO data as a high-resolution tracer of the ASM anticyclone. Figure 7 is a companion one for Figure 6. For water vapor, however, we use MLS data because MLS water vapor measurements in the lower stratosphere have been well validated with e.g., balloon measurements (e.g., Hurst et al., 2016; Fujiwara et al., 2010; Vömel et al., 2007), while reanalysis water vapor data in the lower stratosphere are in general less reliable (e.g., Davis et al., 2017).

Thus, we will make the following revisions:

In Section 2.2, in the second paragraph, we will add the following sentences:
"CAMS CO data are originally provided in mass mixing ratio, kg kg$^{-1}$, which are converted to volume

mixing ratio, ppbv, for this study. It is noted that a quick comparison with MLS Version 4.2 Level 2 CO data (Santee et al., 2017; Livesey et al., 2020) at 400 K isentropic surface (in the form of longitude-time diagram like the one in Section 3.2) shows that CAMS CO data are roughly ~10 ppbv greater than MLS CO over Japan during August–September 2018, but also shows that eastward extension signals coming over Japan agree fairly well qualitatively within the differences in spatio-temporal sampling of the two data sets."

In Section 2.2, we will have the following new (third) paragraph for MLS water vapor data:
"MLS Version 4.2 Level 2 water-vapour data (Santee et al., 2017; Livesey et al., 2020) are analysed because water vapour is also a good tracer of the ASM anticyclone. We use MLS data rather than CAMS data for lower stratospheric water vapour because MLS data have been well validated with e.g., balloon-borne frost-point hygrometers (e.g., Hurst et al., 2016; Fujiwara et al., 2010; Vömel et al., 2007), while reanalysis water vapor data are in general less reliable in the lower stratosphere (e.g., Davis et al., 2017). We found that CAMS water vapour volume mixing ratio data (converted from the original specific humidity data) are greater than MLS data at 400 K isentropic surface over Japan during July–September 2018 (e.g., the differences are roughly ~2 ppmv for the wet signals around the longitudes of Japan in August 2018)."

References (will be added):

Davis, S. M., Hegglin, M. I., Fujiwara, M., Dragani, R., Harada, Y., Kobayashi, C., Long, C., Manney, G. L., Nash, E. R., Potter, G. L., Tegtmeier, S., Wang, T., Wargan, K., and Wright, J. S.: Assessment of upper tropospheric and stratospheric water vapor and ozone in reanalyses as part of S-RIP, Atmos. Chem. Phys., 17, 12743–12778, https://doi.org/10.5194/acp-17-12743-2017, 2017.

Fujiwara, M., Vömel, H., Hasebe, F., Shiotani, M., Ogino, S.-Y., Iwasaki, S., Nishi, N., Shibata, T., Shimizu, K., Nishimoto, E., Valverde-Canossa, J. M., Selkirk, H. B., and Oltmans, S. J.: Seasonal to decadal variations of water vapor in the tropical lower stratosphere observed with balloon-borne cryogenic frostpoint hygrometers, J. Geophys. Res., 115, D18304, https://doi.org/10.1029/2010JD014179, 2010.

Hurst, D. F., Read, W. G., Vömel, H., Selkirk, H. B., Rosenlof, K. H., Davis, S. M., Hall, E. G., Jordan, A. F., and Oltmans, S. J.: Recent divergences in stratospheric water vapor measurements by frost point hygrometers and the Aura Microwave Limb Sounder, Atmos. Meas. Tech., 9, 4447–4457, https://doi.org/10.5194/amt-9-4447-2016, 2016.

Vömel, H., Barnes, J. E., Forno, R. N., Fujiwara, M., Hasebe, F., Iwasaki, S., Kivi, R., Komala, N., Kyrö, E., Leblanc, T., Morel, B., Ogino, S.-Y., Read, W. G., Ryan, S. C., Saraspriya, S., Selkirk, H., Shiotani, M., Valverde Canossa, J., and Whiteman, D. N.: Validation of Aura MLS water vapor by balloonborne Cryogenic Frostpoint Hygrometer measurements, J. Geophys. Res., 112, D24S37, https://doi.org/10.1029/2007JD008698, 2007.

[Figure]

Figure R2-1. (Left) Same as Figure 7 (the revised version). (Right) Same as left but for MLS CO data. Data for the 30°N–40°N region have been aggregated into 3-day and 8°-longitude bins, each constituting about 10 individual data points. The contours for 45 ppbv and 55 ppbv as well as 65 ppbv are added as dotted lines.

[Figure]

Figure R2-2. (Left) Longitude–time distribution of daily averaged water vapor volume mixing ratio at 400 K potential temperature averaged over 30°N–40°N, using CAMS reanalysis specific humidity data. The contour lines for 7, 8, 9, 10, 11 ppmv have been added. (Right) Same as Figure 8 (the revised version).

- L268-269: types of particle and gas --> types of particles and gases

Will be corrected.

- L286: but not have --> but had not

Will be corrected.

- L287: I think it would be appropriate to add "at least not in a monthly mean view" at the end of this sentence (mirroring the zonal mean caveat in L282).

Will be added.

- L288: unlikely due to --> unlikely to be due to

Will be corrected.

**Section 4**
- L311: measurements at --> measurements made at; also, delete the comma after "2018"

Will be corrected.

- L314: the BSR --> BSR

Will be deleted.

- L315-316: The authors might want to specifically note that none of the 11 nights on which data were taken at Fukuoka fell during the intervals of strong enhancement observed at Tsukuba.

We will add to this sentence "and due to the fact that the dates of lidar operation at Fukuoka did not overlap those at Tsukuba when strong enhancement was observed"

- L318: are of a few --> are a few

Will be corrected.

- L319: originate in the ASM anticyclone in association with eastward shedding vortices --> originate in the ASM anticyclone and are transported over these sites in association with eastward shedding vortices

Will be added.

- L320: eruptions and extensive --> eruptions or extensive

Will be corrected.

- L325-330: I'm not convinced that this is the best place for the discussion of the OHP measurements. I think it is generally inappropriate to introduce new aspects in the "Summary and Conclusions" section. In fact, these lines might belong in the Introduction.

This sentence will be moved to the Introduction.

- L332: Add a comma after "(3%–10%)".

Will be added.

- L337: Is it necessary to repeat "PDR" 3 times in this line (i.e., is it needed after "8%" and "4%")?

Will be deleted.

- L340-342: It seems to me that here again the Moana Loa and OHP information in these lines is out of place. Since these data add to the evidence that the signals observed at Tsukuba and Fukuoka must have arisen from the ATAL, this discussion could be moved to Section 3.3, which could then be renamed to reflect its exploration of other potential causes of the observed enhancements and not just focus on "Satellite aerosol data".

We will move these sentences to the end of Section 3.3, whose title will be changed to "Investigation of other potential causes".

- L341: any enhancement --> no enhancement

Will be corrected.

- L360: Delete the comma after "extent"; also, is --> are

Will be corrected.

**References**
- L440: The paper by Hanumanthu et al. has now been accepted for publication in ACP.

Will be updated.

- L473: The reference for the MLS Data Quality Document (Livesey et al.) is missing the year.

The year 2020 will be added.

[Figure]

Revised Figure 1: The colour bar has been slightly changed.

[Figure]

Revised Figure A1: The colour bar has been slightly changed.

[Figure]

Revised Figure 4: The colour code for geopotential height has been changed (to narrow the range of Z). The CO isolines for different months are expressed with different line styles (i.e., dotted for July, solid for August, and dash-dotted for September).

[Figure]

Revised Figure 5: The colour code for geopotential height has been changed (to narrow the range of Z). The CO isolines for different months are expressed with different line styles (i.e., dotted for July, solid for August, and dash-dotted for September).

[Figure]

Revised Figure 6: PV has been replaced with Montgomery streamfunction (MSF; coloured contours at intervals of $0.01 \times 10^5$ m$^2$ s$^{-2}$). Also, CO and MSF data are now instantaneous at 00 UTC, not daily averages.

[Figure]

Revised Figure 7: Coloured line segments have been added for the periods when CO concentration was ≥~60 ppbv along the longitude of Tsukuba (i.e., 3–15, 20–24, and 28–31 August, and 3–8, 14–17, and 28–29 September).

---

## Author Comment (AC3) · 16 Dec 2020

**Response to comments by Anonymous Referee #3**

Thank you very much for your review.

Comments on Manuscript No. ACP-2020-980

In the last decade the Asian Tropopause Aerosol Layer (ATAL) becomes in the focus of attention. This study shows that the transport of aerosol out of the ATAL by eastward shedding vortices were measured over Japan by two lidar systems during summer 2018. Several eddy shedding events were observed and backward trajectory calculations indicate that eddies including air masses with enhanced aerosol particles originate in the Asian monsoon anticyclone. The analysis of satellite observations and meteorological reanalysis confirm the eddy events and further show that the considered time period was free of the impact of volcanic eruptions and high forest fires.

This is a very interesting study, which merits its publication in ACP. The scientific content, the quality of the study and its presentation is good. Therefore, I suggest only some minor revisions before publication by ACP

Thank you very much for your evaluation.

1) P2/L51: 'The enhanced aerosol particle signature in the ASM anticyclone at 14–18 km altitude is known as the Asian Tropopause Aerosol Layer (ATAL), which was believed to consist of carbonaceous and sulphate materials, mineral dust, and nitrate particles (Vernier et al., 2015, 2018; Brunamonti et al., 2018; Bossolasco et al., 2020; Hanumanthu et al., 2020).'

From this statement it is not clear if the knowledge about the chemical composition of the the ATAL particles is based on in situ measurements, remotes sensing observations or model simulations. The discussion about the chemical composition of the ATAL should be much more improved and clarified. I recommend to add a short summery about the current knowledge of ATAL particle characteristics (e.g. chemical composition, particle size distribution, particle form, possible sources etc.). This would help to better bring the results of the lidar measurements presented in this study into the context of previous publications.

We would like to point out that one of the latest ATAL studies by Bossolasco et al. (2020) (a modeling study under review) discuss in the Introduction that "The sources, chemical composition and spatial and temporal variability of the ATAL are not yet well understood." Also, Hanumanth et al. (2020) (a study using balloon borne backscatter sondes) discuss in the Introduction that "The source regions of ATAL aerosols and their chemical precursors on the Earth's surface (origin) as well as the transport pathways from the surface to ATAL altitudes are poorly understood." Note also that Referee #2 suggested us to change from "was believed" to "is believed" in this sentence. Furthermore, in the text, in the following sentences, we introduce the recent study by Höpfner et al. (2019) rather extensively. We believe that this is the current status of our knowledge regarding the chemical composition of ATAL. Thus, we believe

that the current text is not inappropriate for the introduction to a study using ground-based lidar systems.

2) P4/L114: ERA5 is a very new product from ECMWF, therefore I think it is worth to add a few references demonstrating the quality of ERA5 compared to the former ERA-Interim reanalysis.

We will add the following sentences: "ERA5 temperature data in the tropical tropopause layer have been evaluated by Tegtmeier et al. (2020). Lagrangian transport calculations using ERA5 and its predecessor ERA-Interim have been compared by Hoffmann et al. (2019) and Li et al. (2020)."

Tegtmeier, S., Anstey, J., Davis, S., Dragani, R., Harada, Y., Ivanciu, I., Pilch Kedzierski, R., Krüger, K., Legras, B., Long, C., Wang, J. S., Wargan, K., and Wright, J. S.: Temperature and tropopause characteristics from reanalyses data in the tropical tropopause layer, Atmos. Chem. Phys., 20, 753–770, https://doi.org/10.5194/acp-20-753-2020, 2020.

Hoffmann, L., Günther, G., Li, D., Stein, O., Wu, X., Griessbach, S., Heng, Y., Konopka, P., Müller, R., Vogel, B., and Wright, J. S.: From ERA-Interim to ERA5: the considerable impact of ECMWFs next-generation reanalysis on Lagrangian transport simulations, Atmos. Chem. Phys., 19, 3097–3124, https://doi.org/10.5194/acp-19-3097-2019, 2019.

Li, D., Vogel, B., Müller, R., Bian, J., Günther, G., Ploeger, F., Li, Q., Zhang, J., Bai, Z., Vömel, H., and Riese, M.: Dehydration and low ozone in the tropopause layer over the Asian monsoon caused by tropical cyclones: Lagrangian transport calculations using ERA-Interim and ERA5 reanalysis data, Atmos. Chem. Phys., 20, 4133–4152, https://doi.org/10.5194/acp-20-4133-2020, 2020.

3) P4/L122: Please add a statement like this: 'CO and the ATAL have not necessarily the same emission sources, however CO is a good chemical tracer to indicated the location of the Asian monsoon anticyclone.'

We will add the following sentence:
Although CO and ATAL aerosol particles do not necessarily have the same emission sources, CO is a good chemical tracer to indicate the location of the ASM anticyclone.

4) P5/L152: Is it possible that cirrus signal overlays the aerosol signal, so that cirrus and aerosol can coexist simultaneously? Or can you exclude this with your method?

Yes, it is always possible that cirrus signal overlays the aerosol signal if they coexist because of the larger backscattering cross section of cirrus particles. In that case, the BSR and PDR values would become those for cirrus particles.

5) P9/Fig.3: You could add a BSR profile from pre- or post-monsoon to show the difference. The difference can be use to better highlight the signal in BSR from the ATAL.

Figures R3-1 and R3-2 below show the lidar profiles at Tsukuba on some days in May 2018 and in October 2018, respectively. These figures show that BSR in these months did not show enhancements of >1.1 like those found in August and September (Figs. 2c, e). However, we note that TDR slightly increased below 20 km, suggesting a possibility of presence of minute amount of non-spherical particles. The origins of the particles are unknown and a subject of our future study.

Figure R3-3 below shows the lidar profiles at Fukuoka on some days in May-June 2018 and in October-November 2018. Again, we did not detect depolarization enhanced layers (with depolarization ratio higher than 2%) in these months.

Tsukuba 2018/05/02 00:01:51-04:18:47LST
Tsukuba 2018/05/04 19:03:46-00:59:49LST
Tsukuba 2018/05/10 19:04:06-00:59:07LST
Tsukuba 2018/05/13 00:04:52-00:59:41LST
Tsukuba 2018/05/14 19:06:45-00:58:32LST
Tsukuba 2018/05/15 19:07:38-00:59:50LST
Tsukuba 2018/05/18 00:02:54-00:58:03LST
Tsukuba 2018/05/19 19:06:38-00:58:42LST
Tsukuba 2018/05/22 19:08:30-00:58:06LST

Figure R3-1. Lidar profiles taken at Tsukuba on May 2, 4, 10, 13, 14, 15, 18, 19, 22, 24, and 28, 2018. The horizontal dashed line in each panel indicates the location of the first lapse rate tropopause. It is noted that for "TDR*10" = 0.05 means TDR = 0.05/10 = 0.005, i.e., 0.5%TDR. This is close to the depolarization ratio value, 0.366% for air molecules.

Tsukuba 2018/10/02 19:34:54-00:59:27LST
Tsukuba 2018/10/06 19:01:21-00:58:31LST
Tsukuba 2018/10/08 19:19:53-00:59:51LST
Tsukuba 2018/10/09 19:01:14-00:59:24LST
Tsukuba 2018/10/18 19:00:24-00:25:19LST
Tsukuba 2018/10/21 19:01:46-00:57:00LST
Tsukuba 2018/10/24 19:00:27-00:59:22LST
Tsukuba 2018/10/25 19:00:27-00:59:42LST
Tsukuba 2018/10/28 19:03:08-00:59:09LST

[Figure]

Figure R3-2. As for Figure R1-1, but for October 2, 6, 8, 9, 18, 21, 24, 25, 28, 29, and 30, 2018.

[Figure]

Figure R3-3. Lidar profiles taken at Fukuoka on the days in May-June 2018 and in October-November 2018 when the lidar was operated.

6) P10/L204: '... whereas those without enhanced aerosol particles tend to originate from edge regions surrounding the anticyclone.' and from the extratropical lower stratosphere. Right?

At the northern edge regions, yes. But, at the southern edge regions, they are from tropical lower stratosphere.

Why do you use only ten-days backward trajectories? What about 15- or 20-day backward calculations? In somewhat

longer trajectories, the difference between air masses from the core the anticyclone or from the edge (or outside from the anticyclone) should be more pronounced.

Figure R3-4 below shows the 15-day backward trajectories. We had found that if we plot longer trajectories, we have more trajectories within the ASM anticyclone, although the density is still lower for the cases without enhanced aerosols measured at Tsukuba/Fukuoka. Also, we had received a comment before the paper submission that longer trajectories are less reliable. Thus, we decided to show the results from 10-day backward trajectories which better shows the differences.

[Figure]

Figure R3-4. As for Figures 4 (for Tsukuba, for the left two panels) and 5 (for Fukuoka, for the right two panels), but for 15-day backward trajectories. (Note that the colour code for the geopotential height has been revised (to narrow the range of Z); please see the next QA.)

7) P11/Fig.4: Is it possible to adjust the color bar more to the Z range of the trajectories to better highlight the gradients along the trajectories. The bluish colors are only used for one trajectory over the Pacific in Fig. 4b. It looks like that this trajectory is influenced by a tropical cyclone. If that is true that could be mentioned as a side remark.

The colored geopotential height range of Figures 4 and 5 will be changed (narrowed); please see the revised version of these figures at the end of this reply letter.

Regarding the trajectory originating in the middle of the North Pacific at that altitude (Figure 4, bottom panel), we will add the following note in the text: "Note that there is a trajectory that originates in the Pacific south of Japan as low as 4 km (Figure 4, bottom, a small-scale spiral in purple); this is associated with upward transport in the typhoon Soulik."

8) P13/L230 : 'PV can be regarded as a dynamical tracer, with lower values in the ASM anticyclone along the same latitudes (e.g., 30∘N), although background positive gradients in latitude and its noisier nature give more complicated features.'

PV can be very useful to see the edge of the Asian monsoon anticyclone at around 380K (e.g. Ploeger et al., 2015), above around 400K as shown in Fig. 6, the PV is not so useful. Instead you could try to use the (Montgomery) stream function or the geopotential height.

Ploeger, F., Gottschling, C., Grießbach, S., Grooß, J.-U., Günther, G., Konopka, P., Müller, R., Riese, M., Stroh, F., Tao, M., Ungermann, J., Vogel, B., and von Hobe, M.: A potential vorticity-based determination of the transport barrier in the Asian summer monsoon anticyclone, Atmos. Chem. Phys., 15, 13 145–13 159, https://doi.org/doi:10.5194/acp-15-13145-2015, 2015.

Thank you very much for your comment and suggestion. We will change from PV to Montgomery streamfunction. Please see the response letter to Referee #2 for the revised Figure 6. We will add the following sentences in the second paragraph of Section 2.2:

"Carbon monoxide (CO), temperature ($T$), and geopotential ($\Phi$) data are analysed in this paper."

"Montgomery streamfunction (MSF), defined as MSF = $c_p$ $T$ + $\Phi$, where $c_p$ is specific heat of dry air at constant pressure, in isentropic coordinates corresponds to geopotential (height) in pressure coordinates (e.g., Amemiya and Sato, 2018) , and thus is a good dynamical indicator of the ASM anticyclone. Potential vorticity (PV) on isentropic surfaces (e.g., at 360–380 K) is often used as a dynamical tracer in studies of the ASM anticyclone (e.g., Popovic and Plumb, 2001; Garny and Randel, 2013; Ploeger et al., 2015; Amemiya and Sato, 2018); however, PV at and above 400 K (this isentropic surface will be focused in Section 3.2) is not very useful to analyse the ASM anticyclone boundary. Thus, we will analyse MSF at 400 K surface calculated from CAMS data."

Also, we will revise the second paragraph of Section 3.2 accordingly.

9) P16/Fig.8: Why do you show H2O from MLS and not CO from MLS? CO would be a better chemical tracer for transport as H2O which is in addition affected by microphysics. You could also use MLS O3 which should be anticorrelated to CO (low O3 in the anticyclone and high O3 in the lower stratosphere).

Figures R3-5 and R3-6 below show the comparisons of CAMS data and MLS data for CO and for water vapor. As we can see, for both CO and water vapor, CAMS and MLS show qualitatively and broadly similar eastward extension signals over Japan; however, CAMS CO is greater than MLS CO (e.g., the differences are ~10 ppbv around the longitudes of Japan through August–September 2018), and CAMS water vapor mixing ratios are greater than MLS water vapor (e.g., the differences are roughly ~2 ppmv for the wet signals around the longitudes of Japan in August 2018). In this paper, we primarily use CAMS CO data as a high-resolution tracer of the ASM anticyclone. Figure 7 is a companion one for Figure 6. For water vapor, however, we use MLS data because MLS water vapor measurements in the lower stratosphere have been well validated with e.g., balloon measurements (e.g., Hurst et al., 2016; Fujiwara et al., 2010; Vömel et al., 2007), while reanalysis water vapor data in the lower stratosphere are in general less reliable (e.g., Davis et al., 2017).

Thus, we will make the following revisions:

In Section 2.2, in the second paragraph, we will add the following sentences:

"CAMS CO data are originally provided in mass mixing ratio, kg kg$^{-1}$, which are converted to volume mixing ratio, ppbv, for this study. It is noted that a quick comparison with MLS Version 4.2 Level 2 CO data (Santee et al., 2017; Livesey et al., 2020) at 400 K isentropic surface (in the form of longitude-time diagram like the one in Section 3.2) shows that CAMS CO data are roughly ~10 ppbv greater than MLS CO over Japan during August–September 2018, but also shows that eastward extension signals coming over Japan agree fairly well qualitatively within the differences in spatio-temporal sampling of the two data sets."

In Section 2.2, we will have the following new (third) paragraph for MLS water vapor data:

"MLS Version 4.2 Level 2 water-vapour data (Santee et al., 2017; Livesey et al., 2020) are analysed because water vapour is also a good tracer of the ASM anticyclone. We use MLS data rather than CAMS data for lower stratospheric water vapour because MLS data have been well validated with e.g., balloon-borne frost-point hygrometers (e.g., Hurst et al., 2016; Fujiwara et al., 2010; Vömel et al., 2007), while reanalysis water vapor data are in general less reliable in the lower stratosphere (e.g., Davis et al., 2017). We found that CAMS water vapour volume mixing ratio data (converted from the original specific humidity data) are greater than MLS data at 400 K isentropic surface over Japan during July–September 2018 (e.g., the differences are roughly ~2 ppmv for the wet signals around the longitudes of Japan in August 2018)."

References (will be added):

Davis, S. M., Hegglin, M. I., Fujiwara, M., Dragani, R., Harada, Y., Kobayashi, C., Long, C., Manney, G. L., Nash, E. R., Potter, G. L., Tegtmeier, S., Wang, T., Wargan, K., and Wright, J. S.: Assessment of upper tropospheric and stratospheric water vapor and ozone in reanalyses as part of S-RIP, Atmos. Chem. Phys., 17, 12743–12778, https://doi.org/10.5194/acp-17-12743-2017, 2017.

Fujiwara, M., Vömel, H., Hasebe, F., Shiotani, M., Ogino, S.-Y., Iwasaki, S., Nishi, N., Shibata, T., Shimizu, K., Nishimoto, E., Valverde-Canossa, J. M., Selkirk, H. B., and Oltmans, S. J.: Seasonal to decadal variations of water vapor in the tropical lower stratosphere observed with balloon-borne cryogenic frostpoint hygrometers, J. Geophys. Res., 115, D18304, https://doi.org/10.1029/2010JD014179, 2010.

Hurst, D. F., Read, W. G., Vömel, H., Selkirk, H. B., Rosenlof, K. H., Davis, S. M., Hall, E. G., Jordan, A. F., and Oltmans, S. J.: Recent divergences in stratospheric water vapor measurements by frost point hygrometers and the Aura Microwave Limb Sounder, Atmos. Meas. Tech., 9, 4447–4457, https://doi.org/10.5194/amt-9-4447-2016, 2016.

Vömel, H., Barnes, J. E., Forno, R. N., Fujiwara, M., Hasebe, F., Iwasaki, S., Kivi, R., Komala, N., Kyrö, E., Leblanc, T., Morel, B., Ogino, S.-Y., Read, W. G., Ryan, S. C., Saraspriya, S., Selkirk, H., Shiotani, M., Valverde Canossa, J., and Whiteman, D. N.: Validation of Aura MLS water vapor by balloonborne Cryogenic Frostpoint Hygrometer

measurements, J. Geophys. Res., 112, D24S37, https://doi.org/10.1029/2007JD008698, 2007.

[Figure]

Figure R3-5. (Left) Same as Figure 7 (the revised version). (Right) Same as left but for MLS CO data. Data for the 30°N–40°N region have been aggregated into 3-day and 8°-longitude bins, each constituting about 10 individual data points. The contours for 45 ppbv and 55 ppbv as well as 65 ppbv are added as dotted lines.

[Figure]

Figure R3-6. (Left) Longitude–time distribution of daily averaged water vapor volume mixing ratio at 400 K potential temperature averaged over 30°N–40°N, using CAMS reanalysis specific humidity data. The contour lines for 7, 8, 9, 10, 11 ppmv have been added. (Right) Same as Figure 8 (the revised version).

Two figures for ozone from CAMS and MLS are shown below. Figure R3-7 shows horizontal distributions of daily ozone at 400 K during 18–23 August 2018 using CAMS reanalysis data, similar to Figure 6. Figure R3-8 shows longitude–time distribution (Hovmöller diagram) of ozone at 400 K averaged over 30°N–40°N using CAMS reanalysis and MLS data. We see that ozone and CO are anticorrelated around the northern edge of the ASM anticyclone (in other words, along the westerly jet stream). In particular, around 16–19 August, a strong stratosphere-to-troposphere intrusion occurred over Japan, which are clearly observed as a strong ozone enhancement event. Around the southeastern and southern part of the ASM anticyclone, however, the relationship between ozone and CO becomes less clear (i.e., not always clearly anticorrelated); ozone concentration is often still relatively high in CO enhanced regions there. This is probably in part due to the transport from the north (i.e., ozone of stratospheric origin) and in part due to upward transport from Asian countries where ozone is photochemically produced (i.e., ozone of tropospheric origin). Thus, in this paper, we decided not to show ozone results as the interpretation is more complicated than CO and water vapour.

[Figure]

Figure R3-7. As for the revised Figure 6, but for ozone volume mixing ratio (black contours with grey tone, with intervals of 0.1 ppmv) from CAMS reanalysis data. CAMS ozone data are originally provided in mass mixing ratio, kg kg$^{-1}$, which are converted to volume mixing ratio, ppmv, for this response letter.

[Figure]

Figure R3-8. (Left) Same as Figure 7 (the revised version), but for CAMS ozone in ppmv. The contour interval is 0.1 ppmv, with 0.35 ppmv contours added (dotted). (Right) Same as left but for MLS ozone data. Data for the 30°N–40°N region have been aggregated into 3-day and 8°-longitude bins, each constituting about 10 individual data points.

10) P20/L331: 'The PDR values obtained at Tsukuba, i.e., ~5% (3%–10%) suggest that these enhanced particles are solid particles, rather than spherical, liquid H2SO4 particles (PDR ~0%) or cirrus ice particles (PDR > 25%–30%). The observed values may be consistent with those of solid NH4NO3 particles recently suggested by Höpfner et al. (2019).'

Using the particle depolarization ratio, the study shows that the aerosol particles are most likely solid and it is concluded that the aerosol particles possibly contain NH4NO3. In the literature it is discussed that also carbonaceous aerosols, dust, nitrate-containing aerosol, black carbon and organic carbon could contribute to the chemical composition of the ATAL. Can you exclude with your measurements such types of aerosol particles? Please clarify this point.

The lidar measurements cannot exclude the co-existence/existence of other types of aerosol particles. We will add the following sentence: "However, it should be noted that the lidar BSR and PDR measurements cannot exclude the possibility of co-existence of other types of solid aerosol particles such as mineral dust, black carbon, and some types of carbonaceous aerosols which are solid."

Minor comments:

1) P3/L84: (senkrecht in German) –> (= "senkrecht" in German) ?

Will be added.

2) P5/L37: remove large white spaces

This will be handled at the type-setting phase (if this paper is accepted).

3) P13/L223: 'Horizontal distributions of CO and PV' add 'from CAMS'

Will be added.

4) P13/L233: 'are shown in Figure 7' -> 'are shown as Hovmöller diagrams in Fig. 7'

We chose not to use the term "Hovmöller diagram" because this may not be understandable for some of the readers of the journal Atmospheric Chemistry and Physics. But, we are happy to make the change if the editor recommends to do so.

5) Fig.7/8: You could say that the Figures are 'Hovmöller diagrams'

Please see above.

[Figure]

Revised Figure 4: The colour code for geopotential height has been changed (to narrow the range of Z). The CO isolines for different months are expressed with different line styles (i.e., dotted for July, solid for August, and dash-dotted for September).

[Figure]

Revised Figure 5: The colour code for geopotential height has been changed (to narrow the range of Z). The CO isolines for different months are expressed with different line styles (i.e., dotted for July, solid for August, and dash-dotted for September).

---

## Author Response (AR2)

**Responses to Referee #2 (Dr. Michelle Santee)**

The authors have done a good job in revising the manuscript in response to the comments of the three referees. In my opinion the reviewers' concerns have for the most part been adequately addressed. I have only a few remaining suggestions (mostly minor wording changes where edits or additions were made during revision) that I feel should be considered before publication.

Thank you very much for your evaluation.

• L108-109: Assuming that I am interpreting this sentence correctly, then I think that both instances of "with" in "… uncertainty of the normalization value of BSR with … and that of the extinction-to-backscatter ratio with …" should be "to be". In addition, "uncertainty of BSR were then estimated" should be "uncertainty of BSR was then estimated".

Corrected.

• L140-144: I think it would be good to add "(not shown)" somewhere in this sentence (perhaps right after "quick comparison").

Added.

• L146-148: Since the papers by Popovic & Plumb [2001] and Santee et al. [2017] are already referenced in this manuscript, it would be appropriate to note here that those studies also employed MSF.

These two papers have been added here.

• L151: this isentropic surface will be focused in Section 3.2 --> the isentropic surface we will focus on in Section 3.2

Corrected.

• L158: We found that CAMS --> We found (not shown) that CAMS

Added.

• L181: prevent from characterizing the --> prevent characterization of the

Corrected.

• L182-183: highly variable around this date and was located at 17 km on that date --> highly variable at the time and was located at 17 km on that date

Corrected.

• L196: for Tsukuba data we do not plot the data --> for Tsukuba we do not plot the data

Corrected.

• L287: 20-25 August event--> 20-24 August event

Corrected.

• L325-326: My apologies for not commenting on this point in the original manuscript, but it would be appropriate to add two recent publications on the Australian New Year's pyroCbs to the list of papers showing that extensive wildfires influence stratospheric aerosol loading: Kablick et al. [GRL, 2020; https://doi.org/10.1029/2020GL088101] and Khaykin et al. [Communications Earth & Environment, 2020; https://doi.org/10.1038/s43247-020-00022-5].

Thank you for pointing us to these papers. We have added these.

• L393: showed the PDR values --> showed PDR values

Corrected.

• L443-445: An author was added in revision, but the author contributions section has not been updated accordingly.

Updated.

**Responses to Referee #3**

The manuscript 'Lower-stratospheric aerosol measurements in eastward shedding vortices over Japan from the Asian summer monsoon anticyclone during the summer of 2018' by Fujiwara et al., is ready for publication after revising the following comments:

Thank you for your evaluation.

1) P2/L51: 'The enhanced aerosol particle signature in the ASM anticyclone at 14–18 km altitude is known as the Asian Tropopause Aerosol Layer (ATAL), which is believed to consist of carbonaceous and sulphate materials, mineral dust, and nitrate particles (Vernier et al., 2015, 2018; Brunamonti et al., 2018; Bossolasco et al., 2020; Hanumanthu et al., 2020).'

I still recommend to revise this sentence to better make clear which information is from measurements and which from model simulations (e.g. as follows):

The enhanced aerosol particle signature in the ASM anticyclone at 14–18 km altitude found in satellite as well as in in situ balloon-borne measurements is known as the Asian Tropopause Aerosol Layer (ATAL) (e.g. Vernier et al., 2015, 2018; Brunamonti et al., 2018; Hanumanthu et al., 2020). Based on model simulations ATAL is believed to consist of carbonaceous and sulphate materials, mineral dust, and nitrate particles (e.g. Fadnavis et al., 2013; Gu et al.,2016; Lau et al., 2018; Fairlie et al., 2020; Bossolasco et al., 2020;...). However, only limited information on the chemical composition of the ATAL particles is available from measurements (e.g. Martinsson et al., 2014; Vernier et al., 2018; Höpfner et al., 2019;....).

I just mentioned a few references. Feel free to add some other references or select some other references. I propose to better check the current status of publications to the issue of chemical composition of ATAL.

Fadnavis, S., Semeniuk, K., Pozzoli, L., Schultz, M. G., Ghude, S. D., Das, S., and Kakatkar, R.: Transport of aerosols into the UTLS and their impact on the Asian monsoon region as seen in a global model simulation, Atmos. Chem. Phys., 13, 8771–8786, https://doi.org/10.5194/acp-13-8771-2013, 2013.

Martinsson, B. G., Friberg, J., Andersson, S. M., Weigelt, A., Hermann, M., Assmann, D., Voigtländer, J., Brenninkmeijer, C. A. M., van Velthoven, P. J. F., and Zahn, A.: Comparison between CARIBIC Aerosol Samples Analysed by Accelerator-Based Methods and Optical Particle Counter Measurements, Atmos. Meas. Tech., 7, 2581–2596, https://doi.org/10.5194/amt-7-2581-2014, 2014.

Gu, Y., Liao, H., and Bian, J.: Summertime nitrate aerosol in the upper troposphere and lower stratosphere

over the Tibetan Plateau and the South Asian summer monsoon region, Atmos. Chem. Phys., 16, 6641–6663, https://doi.org/10.5194/acp-16-6641-2016, 2016.

Lau, W. K. M., Yuan, C., and Li, Z.: Origin, Maintenance and Variability of the Asian Tropopause Aerosol Layer (ATAL): The Roles of Monsoon Dynamics, Sci. Rep., 8, 3960, https://doi.org/10.1038/s41598-018-22267-z, 2018.

Fairlie, T. D., Liu, H., Vernier, J.-P., Campuzano-Jost, P., Jimenez, J. L., Jo, D. S., Zhang, B., Natarajan, M., Avery, M. A., and Huey, G.: Estimates of Regional Source Contributions to the Asian Tropopause Aerosol Layer Using a Chemical Transport Model, J. Geophys. Res., 125, e2019JD031506, https://doi.org/10.1029/2019JD031506, 2020.

Thank you very much for the detailed suggestions. We have added the suggested sentences and the references with some modifications as:

The enhanced aerosol particle signature in the ASM anticyclone at 14–18 km altitude was first discovered from satellite observations (Vernier et al., 2011) and thereafter referred to as the Asian Tropopause Aerosol Layer (ATAL). It was later verified from in situ balloon-borne measurements (Vernier et al., 2015, 2018; Yu et al., 2017; Brunamonti et al., 2018; Hanumanthu et al., 2020). Information on the chemical composition of the ATAL particles is limited (e.g. Martinsson et al., 2014; Vernier et al., 2018; Höpfner et al., 2019). Based on model simulations, the ATAL is expected to consist of carbonaceous and sulphate materials, mineral dust, and nitrate particles (e.g., Fadnavis et al., 2013; Gu et al., 2016; Lau et al., 2018; Fairlie et al., 2020; Bossolasco et al., 2020). Through analysis of satellite and high-altitude aircraft observations and laboratory experiments, Höpfner et al. (2019) provided evidence that. . .

2) P22/L400:

'However, it should be noted that the lidar BSR and PDR measurements cannot exclude the possibility of co-existence of other types of solid aerosol particles such as mineral dust, black carbon, and some types of carbonaceous aerosols which are solid.'

Please add here some references for publications that propose that ATAL could consist of solid aerosol particles such as mineral dust, black carbon, and some types of carbonaceous aerosols which are solid. It would be an added value for the paper to know if these references are based on simulations or measurements of ATAL.

We have added some (recent) references as:

". . . the possibility of co-existence of other types of solid aerosol particles such as mineral dust (e.g., modelling work by Lau et al., 2018; in situ measurements by Vernier et al., 2018), black carbon (e.g., modelling work by Gu et al., 2016), and some types of carbonaceous aerosols (e.g., modelling works by Gu et al., 2016; Lau et al., 2018; Fairlie et al., 2020) which are solid."